# Structural basis for prostaglandin and drug transport via SLCO2A1

Chitra Joshi [1,2,8], Justin C. Deme [3,8], Yoshinobu Nakamura [4,5,8], Wei-Tse Hsu[1,6,8], Jonathan D. Goult[1,2], Takafumi Kato[1,2], Joanne L. Parker [1,2], Philip C. Biggin [1,6] ✉, Susan M. Lea [3,7] ✉, Takeo Nakanishi [4] ✉ & Simon Newstead [1,2] ✉

Organic anion-transporting polypeptide transporters (SLCO/OATPs) function as cellular gatekeepers, regulating intestinal absorption, hepatic and renal clearance, and the tissue distribution of drugs and metabolites in the human body. However, the mechanisms underlying substrate selection within the SLCO superfamily remain unclear, hampering efforts to rationalize the interaction of drugs and metabolites with these transporters. SLCO2A1 (also known as OATP2A1) is responsible for the distribution of eicosanoids, including prostaglandins (PGs) and thromboxanes, throughout the body, in addition to several families of nonsteroidal anti-inflammatory drugs (NSAIDs). Here, we present cryogenic electron microscopy structures of SLCO2A1 bound to endogenous PGs and to four widely prescribed medications for treating inflammation, chronic asthma, and Parkinson's disease (PD). Complementary molecular dynamics and in vivo cellular assays elucidate the molecular basis for PG and drug recognition. Our study reports essential mechanistic details that underpin substrate selection and subfamily adaptation within the broader SLCO superfamily of drug and metabolite transporters.

Solute carriers (SLCs) play a crucial role in regulating the uptake and elimination of metabolites, their breakdown products, and exogenous drug molecules across cellular membranes[1–4]. Many FDA-approved drugs interact with plasma membrane SLCs, particularly the OATP and SLC22 families of organic anion and cation transporters[5,6]. Drug-SLC interactions can disrupt or modulate endogenous metabolite levels, potentially leading to adverse side effects or presenting new opportunities for modulating SLCs involved in disease[1,7]. However, the mechanisms controlling substrate and drug selection within the OATP and SLC22 families of SLCs remain unclear.

SLCO2A1, also known as OATP2A1 and SLC21A2[8], is widely expressed throughout the human body and facilitates the uptake of prostaglandins (PGs) and their synthetic analogues across the plasma membrane[5,9,10]. Prostaglandins form part of the larger eicosanoid family of fatty acids, including thromboxanes and leukotrienes. These hormone-like lipids are potent signalling molecules and play essential homeostatic functions in the body, including regulating inflammatory responses[11,12], nociceptive pain perception[13], inducing fever[14], and organ development[15]. In the clinic, synthetic PGs are routinely administered in obstetrics to induce labour and manage postpartum haemorrhage[16], in ophthalmology to treat glaucoma by reducing intraocular pressure[17], and in certain cardiovascular conditions, such as pulmonary hypertension[18]. Although the physiological role of eicosanoids is well documented and their respective

[1]Department of Biochemistry, University of Oxford, Oxford, UK. [2]Kavli Institute of Nanoscience Discovery, University of Oxford, Oxford, UK. [3]Center for Structural Biology, Center for Cancer Research, National Cancer Institute, National Institutes of Health, Frederick, MD, USA. [4]Faculty of Pharmacy, Takasaki University of Health and Welfare, Takasaki, Gunma, Japan. [5]Gunma University Graduate School of Medicine, Maebashi, Gunma, Japan. [6]Structural Bioinformatics and Computational Biochemistry, University of Oxford, Oxford, UK. [7]Structural Biology, St Jude Children's Research Hospital, Memphis, TN, USA. [8]These authors contributed equally: Chitra Joshi, Justin C. Deme, Yoshinobu Nakamura, Wei-Tse Hsu. ✉e-mail: Philip.biggin@bioch.ox.ac.uk; susan.lea@stjude.org; nakanishi@takasaki-u.ac.jp; simon.newstead@bioch.ox.ac.uk

receptors are targeted for therapeutic benefit, the mechanistic details underlying eicosanoid transport in the body remain poorly understood, hampering efforts to develop new medications that modulate eicosanoid signalling.

Prostaglandins are unsaturated carboxylic acids that consist of a 20-carbon skeleton featuring a five-membered ring known as the E-ring and two acyl chains referred to as the ω-chain and the α-chain, the latter of which contains the terminal carboxylic acid group. There are four types of PGs in the human body: $I_2$, $D_2$, $E_2$, and $F_{2\alpha}$, which differ in the placement of hydroxyl and ketone groups within the molecule[19]. PGs are synthesized from arachidonic acid through the actions of cyclooxygenases (COX-1 and COX-2) and exert their effects by activating specific G-protein coupled prostaglandin receptor pathways[20]. SLCO2A1 plays an important role in terminating PG-induced signalling by clearing PGs from their site of action[21–23].

Recent genome-wide association studies (GWAS) have shown that recessive mutations in SLCO2A1 result in abnormal catabolism of $PGE_2$, leading to primary hypertrophic osteoarthropathy (PHO), also known as pachydermoperiostosis (PDP)[24]. Furthermore, dysregulation of SLCO2A1 has been associated with chronic enteropathy and Crohn's disease[25], and pathologically elevated $PGE_2$ levels in the intestine may contribute to the formation of ulcers[26]. From a clinical perspective, inhibitors of SLCO2A1 are currently being studied as a therapy for diabetic foot ulcers[27]. However, without a mechanistic understanding of PG recognition, targeting and modulating SLCO2A1 faces significant challenges[5].

SLCO2A1 is a member of the organic anion transporting polypeptide (OATP/SLCO) family of secondary active transporters[5,9] and part of the Major Facilitator Superfamily (MFS)[28] (Supplementary Fig. 1). Structural insights into SLCO2A1 would address many fundamental questions concerning lipid transport via organic anion transporters and accelerate initiatives targeting PG metabolic pathways for drug development[10]. In this study, we present high-resolution structures of wild-type rat Slco2a1 (lowercase for rat, uppercase for human) bound to endogenous PGs, prostaglandin $E_2$ ($PGE_2$) and prostaglandin $F_{2\alpha}$ ($PGF_{2\alpha}$), which reveal how endogenous PGs are recognised and transported. Additional structures in complex with Zafirlukast, a leukotriene receptor antagonist used in the treatment of chronic asthma[29], and Losartan, an angiotensin receptor blocker prescribed for the treatment of hypertension, reveal how drugs utilize the lipid binding site to gain access to the cell via SLCO2A1. Our results also reveal that Fentiazac, a nonsteroidal anti-inflammatory drug (NSAID)[30] and Tolcapone, an anti-Parkinson's Disease drug[31], are inhibitors of SLCO2A1. The specific inhibition of SLCO transporters is a growing area of drug development and therapeutic strategies[32]. Although several therapeutic drugs and drug classes are known to inhibit organic anion transporters, particularly SLCO1B1 and SLCO1B3, no clear rationale has emerged to guide targeted inhibition of specific members of the SLCO superfamily[33]. The structure of Slco2a1 in complex with these drugs now provides insights into how to develop more selective inhibitors of PG transport.

Together, these structures and associated functional data reveal the architectural features of ligand, drug and inhibitor recognition in SLCO2A1. Interestingly, several of these details differ from the current structures of organic anion transporters (OAT/OATP; SLC22/SLC21)[34–39], highlighting key differences within the OAT/OATP superfamilies and revealing a distinct mechanism for PG uptake into the cell via the lipid membrane.

## Results

### Cryo-EM structure of Slco2a1

To understand the structural basis for PG recognition, we determined the cryogenic electron microscopy (cryo-EM) structure of rat Slco2a1 bound to $PGE_2$ (dinoprostone) at 3.2 Å (Fig. 1, Supplementary

Fig. 2 and Supplementary Table 1) and $PGF_{2\alpha}$ (dinoprost) at 3.4 Å (Supplementary Figs. 3-4a and Supplementary Table 1). Prostaglandin E2 has a ketone group at the C9 position of the cyclopentane ring, which characterises this molecule as a prostaglandin of the 'E' series. Prostaglandin $F_{2\alpha}$, in contrast, has a hydroxyl group at the C9 position of the cyclopentane ring, in addition to a hydroxyl at the C11 position. The designation 'F' and suffix 'α' indicate that the C9 hydroxyl group is in the alpha configuration, meaning it is on the same side of the ring as the C11 hydroxyl group.

Rat Slco2a1 shares 83 % sequence identity (89 % similarity) with the human ortholog[8]. Slco2a1 adopts an outward-open conformation, with the canonical two six-transmembrane (TM) N- and C-terminal bundles of the MFS fold tightly packed on the intracellular side of the membrane. The two 6-TM bundles fan apart towards the extracellular side of the membrane, enclosing a large polar cavity that extends towards the cytoplasm. Consistent with previous structures of SLCO family members[38–42], Slco2a1 contains a large extracellular domain (ECD) inserted between TM 9 and TM 10, resembling a Kazal-like domain consisting of an alpha-beta fold with three conserved disulfide bonds. The ECD features several conserved N-glycosylation sites, which are essential for correctly localizing SLCO2A1 to the plasma membrane[43]. Additional extracellular loops are situated on the N-terminal bundle, between TM 3 and TM 4, and between TM 5 and TM 6, which remain unresolved in our maps. On the C-terminal bundle, we observe another well-resolved extracellular loop between TM 11 and TM 12, which we term the C-loop, that forms an unstructured hairpin that interacts with the membrane-facing side of the Kazal-like ECD. Both structural features are conserved within the broader SLCO/OATP superfamily; however, their roles remain unclear. Finally, the intracellular side of the transporter is characterized by an extension of TM 12, which protrudes beyond the membrane plane and into the cytoplasm, along with an extension of TM 7 that wraps around the intracellular side of the transporter and is called the intracellular helix (ICH)[39,44].

### Recognition of prostaglandin $E_2$ and $F_2\alpha$

$PGE_2$ and $PGF_{2\alpha}$ are located at the base of the central cavity and adopt near identical binding poses, consistent with their comparable affinities for rat Slco2a1 (Ki values of ~25–43 nM[45–47]). The binding pose of $PGE_2$ is also observed in the recent structure of human SLCO2A1[41]. The E-ring in $PGE_2$ and $PGF_{2\alpha}$ is positioned within ~4 Å of a conserved cluster of residues on TM 11, which consists of Arg561, Trp565 and Phe557 (Fig. 1b, c & Supplementary Fig. 4a). We termed this part of the transporter the major binding site, given that this is where the majority of the PG molecule sits within the transporter. Alanine substitutions of these residues cannot transport $PGE_2$, indicating their importance for function in both rat and human SLCO2A1 (Fig. 1d & Supplementary Fig. 4b). Arginine 561 is strictly conserved within the wider SLCO family of organic anion transporters[48–50], while Trp565 and Phe557 are only found in members that recognize PG substrates (SLCO2B1 & SLCO3A1) (Supplementary Fig. 1), indicating these residues may be necessary for PG selectivity. Substitution of Arg561 with alanine abolished transport. Replacement with either lysine, glutamine or leucine reduced activity in human SLCO2A1 to <20 % of WT levels, demonstrating the critical importance of the positively charged guanidino group for PG induced transport. Substitution of Trp565 with alanine also abolished transport, whereas a conservative substitution with phenylalanine retained 40 % of WT transport levels. Replacement with leucine, which is also hydrophobic, abolished transport to <20 % of WT levels. Mutation of Trp565 to glycine has also been observed in patients with pachydermoperiostosis (PDP)[51], demonstrating the importance of this tryptophan to the physiological function of SLCO2A1. Similarly, substitution of Phe557 with alanine abolished activity, whereas substitution with tyrosine retained 60% WT levels.

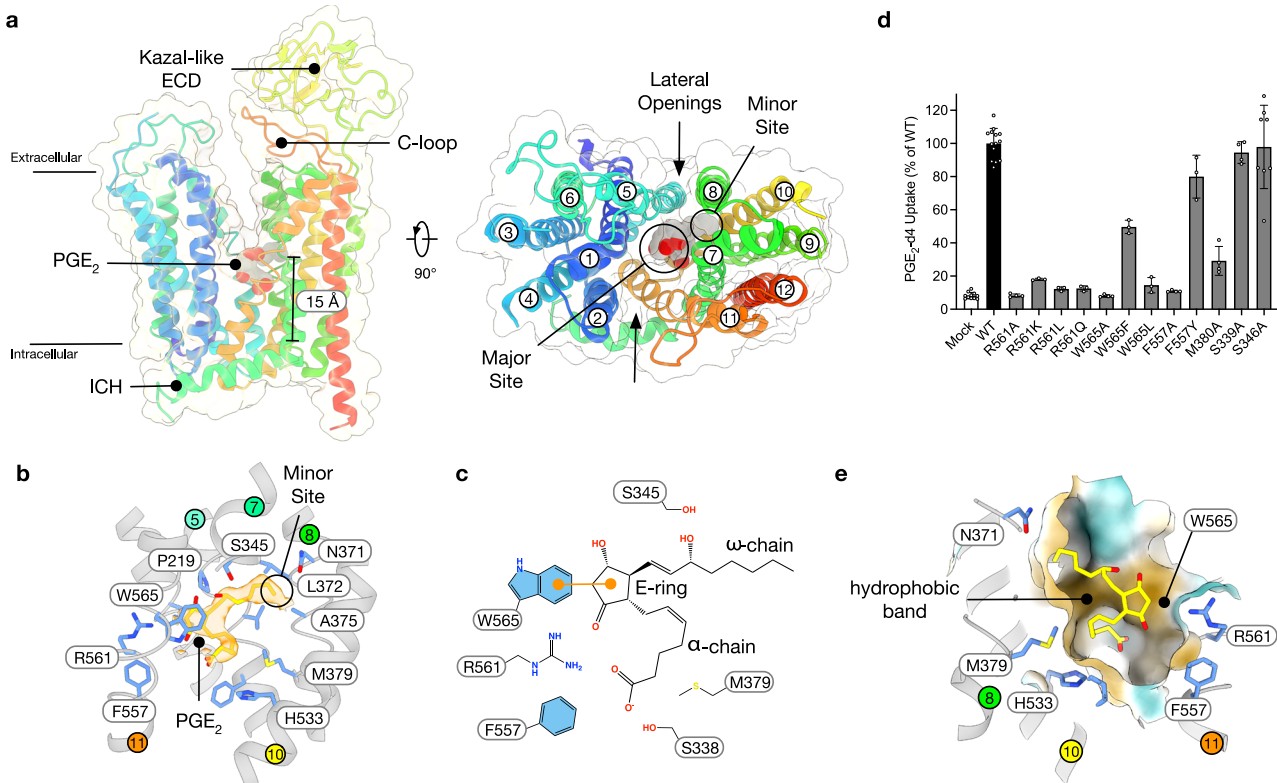

**Fig. 1 | Cryo-EM structure of Slco2a1 bound to PGE$_2$. a** The Cryo-EM structure of Slco2a1 showing the location of the bound PGE$_2$ ligand, along with the Kazal-domain, C-loop, and intracellular helix (ICH) domains. The major and minor sites within the substrate binding cavity and the lateral openings into the membrane bilayer are indicated. **b** A zoomed-in view of the binding site reveals the main interactions formed with PGE$_2$. Key residues that interact with or are near the substrate are displayed as sticks. Cryo-EM volume density for PGE$_2$ is shown (orange and threshold 0.2). **c** Schematic of PGE$_2$ binding interactions. **d** Cell-based transport assays for wild type (WT) and mutant human SLCO2A1. $n = 14$ independent experiments for the WT, $n = 11$ for the mock-transfected control and $n = 4$ (R561A); $n = 3$ (R561K; R561L; R561Q; W565F; W565L; F557Y); $n = 4$ (W565A; F557A; M380A; S339A); $n = 8$ S346A; data shown are the mean and error bars are s.d. **e** A zoomed-in view of the PGE$_2$ binding site, rotated 180° relative to panel b, with the surface coloured by hydrophobicity. The prominent hydrophobic band is indicated along with key interacting residues.

Combined, these results demonstrate the importance of Phe557, Arg561 and Trp565 to SLCO2A1 function and PG transport[41,42].

The carboxylate groups of the α-chain of PGE$_2$ and PGF$_{2α}$, which impart a net negative charge to these molecules at physiological pH, curl beneath their respective E-rings and face the positively charged region created by the Arg561, Phe557, and Trp565 cluster. The position of the carboxylate group in PGE$_2$ and PGF$_{2α}$ may also explain why the aromatic groups of Trp565 and Phe557 are important for PG recognition. The proximity of these two aromatic side chains to Arg561 amplifies the positive charge of this side chain within the binding site, which would facilitate the orientation of the E-ring and carboxylate group observed in the cryo-EM structures. As discussed below, Arg561 is also likely to play an essential role in the general alternating-access mechanism within the SLCO superfamily, as well as in its interaction with Glu78 on TM5.

The positioning of the α-chain beneath the E-ring is also aided by the proximity of Met379 on TM8 (~3.6 Å), which pushes the C1-carboxylate group under the E-ring in both substrates. In PGE$_2$, the C1-carboxylate group forms hydrogen bonds with the NE1 nitrogen on Trp565 on TM11 and Ser338 on TM7, and it is located close (4 Å) to His533 on TM10. Alanine substitution of Met379 (Met380 in human SLCO2A1) also reduces PGE$_2$ transport to ~30 % WT levels, demonstrating the importance of this side chain in PG recognition. However, replacement of Ser338 (Ser339 in human SLCO2A1) had little impact on SLCO2A1 activity (Fig. 1d and Supplementary Fig. 4b), suggesting this side chain does not play an important role in PG

recognition. In contrast, mutation of His533 to glutamate abolishes the transport activity of human SLCO2A1[42].

In PGF$_{2α}$ the interaction profile with the transporter is very similar, except Arg561 now projects into the binding site and sits ~4 Å from the C1-carboxylate (Supplementary Fig. 4a, c), indicating this side chain plays an active role in ligand recognition. Indeed, during 1 μs molecular dynamics (MD) simulations of this structure, which are discussed further below, Arg561, along with Trp565 and Ser338, directly interact with the C1-carboxylate group (Supplementary Fig. 4d-e). In contrast, the ω-chain on PGE$_2$ and PGF$_{2α}$ extends away from their respective E-rings and projects into a hydrophobic cavity formed by Val216 and Pro219 on TM5, Ile341, Leu344 and Ser345 on TM7, Leu372 and Ala375 on TM8, as well as Ala526 and Cys530 on TM 10. We labelled this the minor site. The hydroxyl group on the ω-chain in PGE$_2$ is situated within hydrogen bonding distance to Ser345 (3.4 Å) and the E-ring hydroxyl at C11 (2.8 Å), which may further stabilize the lipid in this orientation (Fig. 1b, c). Substitution of Ser345 (Ser346 in human SLCO2A1) with alanine reduced PGE$_2$ transport to 80% WT levels (Fig. 1d), indicating a subtle, but measurable, effect of PG recognition.

A notable feature of the binding poses for both PG molecules is their location relative to a prominent hydrophobic band in the binding pocket (Fig. 1e & Supplementary Fig. 4c). The E-rings are oriented towards the positive charge on Arg561 and are positioned adjacent to the conserved side chain Trp565, which forms the beginning of this hydrophobic band. The α- and ω-chains extend above and below the plane of this hydrophobic band, which also lies at the centre of the transporter. As discussed below, the position of

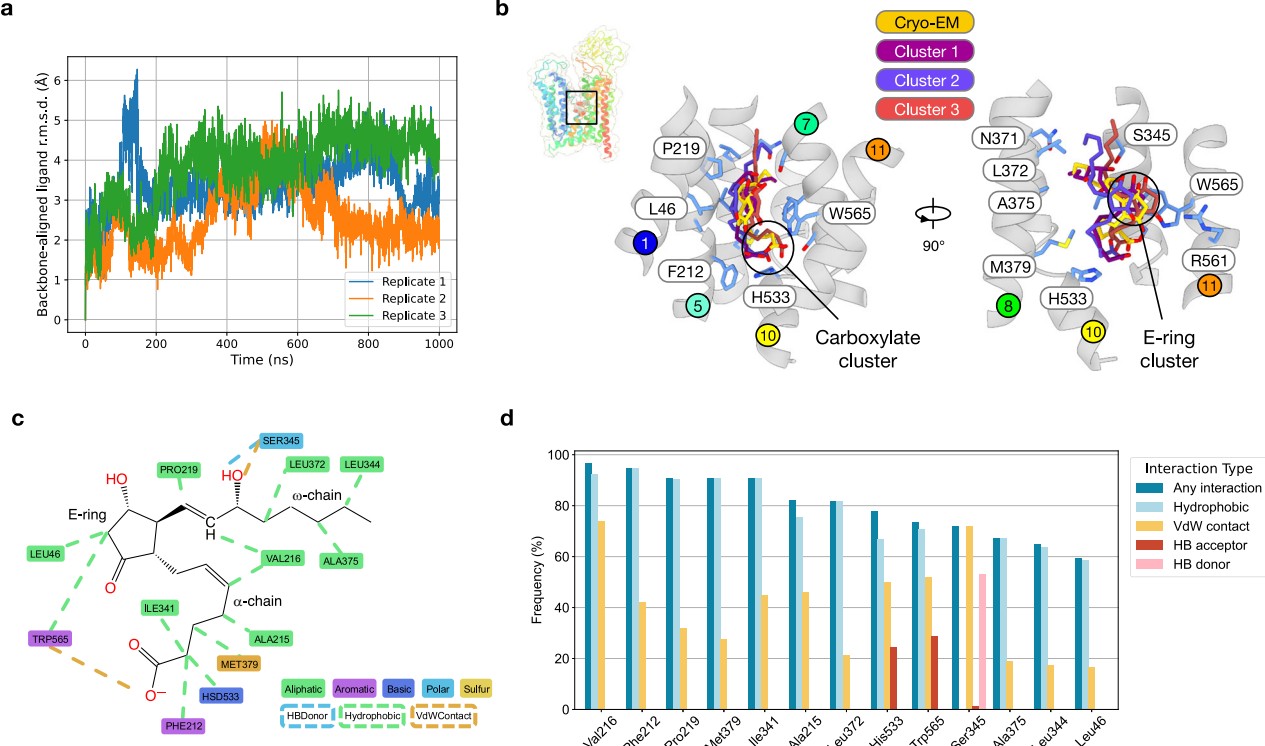

**Fig. 2 | Molecular dynamics analysis of PG interactions. a** A 2D plot showing backbone-aligned ligand r.m.s.d values of the PGE$_2$ position in three replicates of the 1µs simulation. **b** A zoomed-in view of the binding site shows the three most representative binding structures obtained by clustering the trajectory pooled across three microsecond-long unbiased molecular dynamics simulation replicates. The starting pose was the cryo-EM structure of Slco2a1 bound to PGE$_2$ (inset left). The key interacting residues within <6 Å of the bound PG molecule are highlighted. **c** An interaction map of the residues within 10 Å of the PGE$_2$ molecule from the three replicates of the simulation. Only interactions present in more than 50 % of the simulation frames are shown. **d** A 2D plot showing the frequency of different types of interactions observed for residues that interacted with the ligand in more than 50 % of the simulation frames. Results were calculated from the trajectory pooled across all three replicates.

this band serves as the pivot point in the structural rearrangement of the transporter between outward- and inward-facing states, thereby placing PG molecules in an optimal position for alternating-access transport across the membrane.

**Analysis of prostaglandin recognition using molecular dynamics**
To gain further insight into the mechanism of PG recognition, we undertook unbiased MD simulations of rat Slco2a1 in complex with PGE$_2$ and PGF$_{2\alpha}$ using the cryo-EM structures as starting poses. We conducted three 1 µs simulations to assess the stability of PGE$_2$ and PGF$_{2\alpha}$ in the binding site (Fig. 2a & Supplementary Fig. 4f-g). Clustering analysis of the MD trajectories demonstrates that PGE$_2$ and PGF$_{2\alpha}$ remain reasonably stable in their respective binding poses, with ligand root mean square deviation (r.m.s.d) values averaged over all trajectories of 3.26 Å ± 0.96 for PGE$_2$ and 4.23 Å ± 1.10 for PGF$_{2\alpha}$, respectively. The ligand r.m.s.d values compare favourably with those calculated from the protein backbone, which were in the range of 2.80–6.97 Å, indicating the ligand flexibility is consistent with the overall flexibility of the protein. The modelled positions of the ligands are further supported by the low number of clusters (12 for PGE$_2$ and 17 for PGF$_{2\alpha}$) and by the fact that the three largest clusters accounted for 95% and 77% of the simulation frames, respectively. A striking result from the cluster analysis is the stability of both the C1-carboxylate and the E-ring in the simulations (Fig. 2b), suggesting that these are important features determining ligand recognition and selection within SLCO2A1.

We next analyzed which residues interacted most frequently with PGE$_2$ and PGF$_{2\alpha}$. Residues that interacted > 50 % of the time with PGE$_2$ are located on TM 5, TM 7 and TM 8 and cluster around the ω-chain (Fig. 2c). In the PGF$_{2\alpha}$ simulations, the majority of interactions are

made to the α-chain, with far fewer to the ω-chain (Supplementary Fig. 4e). This may explain the higher r.m.s.d. values for this PGF$_{2\alpha}$ compared to PGE$_2$ and why the ω-chain may play a less important role in ligand selection. Tryptophan 565 is unique to SLCO family members that are dedicated PG transporters and interacts with the C1-carboxylate on the α-chain in both PGE$_2$ and PGF$_{2\alpha}$. In the PGF$_{2\alpha}$ bound structure, Arg561 and Ser338 also interact with the C1-carboxylate group (Supplementary Fig. 4f). Further analysis of MD simulations reveals a dominant role for van der Waals contacts and hydrophobic interactions along with hydrogen bonds to Ser345, Ser338, His533, Arg561 and Trp565 (Supplementary Fig. 4e-f) in binding the PG molecules.

Overall, the MD simulations align with our mutagenesis data (Fig. 1d, Supplementary Fig. 4b) and highlight the role of Trp565 and hydrophobic interactions in orienting the PG ligands.

**Structural basis for drug recognition by SLCO2A1**
SLCO2A1 is noted for its interactions with several commonly prescribed drug classes, such as NSAIDs, antihypertensives, anti-PD medications, and asthma treatments[33]. As direct uptake is often not measured, whether these drugs function as inhibitors or are transported remains unclear for many members of the SLCO superfamily. However, using mass spectrometry, we measured the ability of human SLCO2A1 to transport representative drugs from these four drug families. Our results show that Zafirlukast, the anti-leukotriene receptor antagonist, and Losartan, an angiotensin receptor blocker used to treat hypertension, are transported (Fig. 3a). In contrast, Fentiazac, an NSAID, and Tolcapone, used to treat PD, showed no detectable uptake. The calculated IC$_{50}$ values for Zafirlukast, Losartan, Fentiazac and

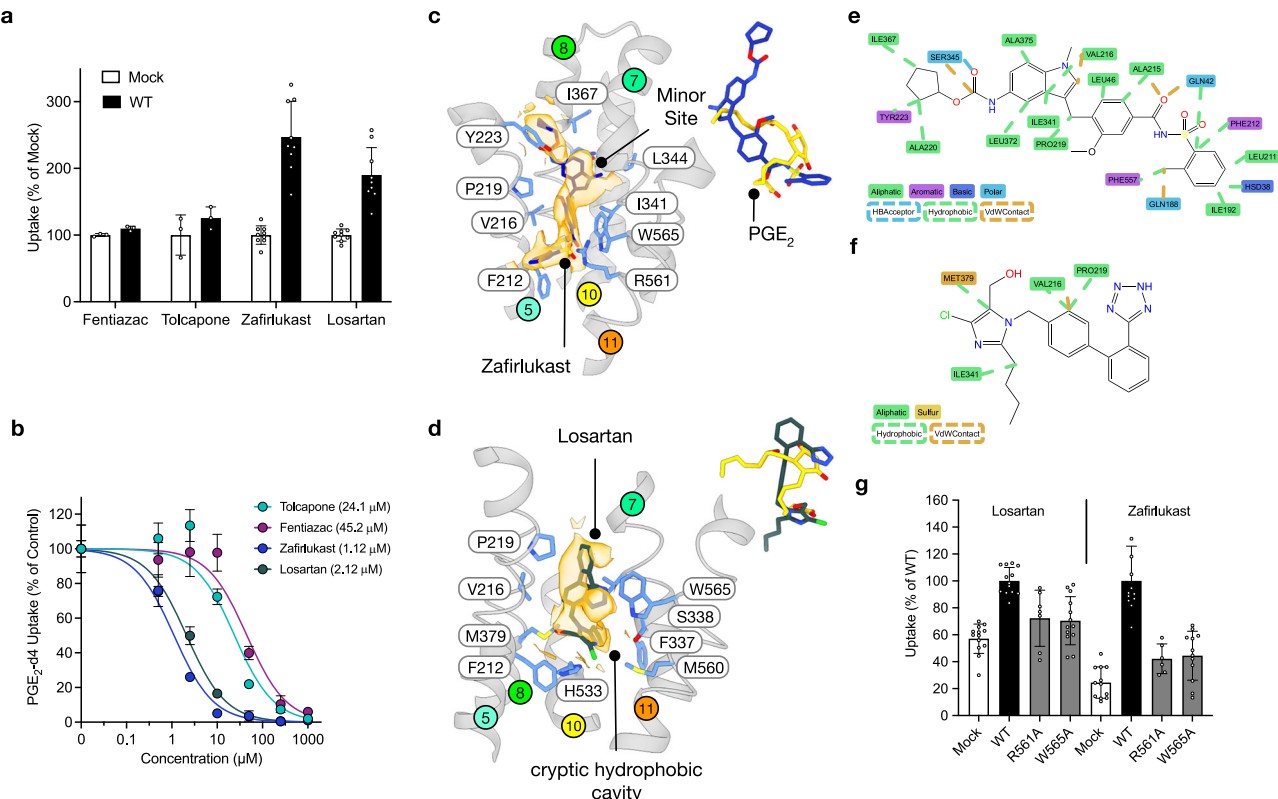

**Fig. 3 | Drug transport and recognition by Slco2a1. a** Cell-based transport assays of human SLCO2A1 with four clinically prescribed drug molecules. $n = 3$ independent experiments (Fentiazac and Tolcapone) and $n = 9$ independent experiments (Zafirlukast and Losartan); data shown are the mean and error bars are s.d. **b** IC$_{50}$ data for the four drug molecules; errors are s.d. **c** A zoomed-in view of the binding site shows the position of Zafirlukast. Cryo-EM volume density is shown (orange and threshold 0.2). Inset—Structural overlay with the bound PGE$_2$ molecule (PDB: 9R0M). **d** A zoomed-in view of the binding site shows the position of Losartan.

Cryo-EM volume density is shown (orange and threshold 0.2). Inset—Structural overlay with the bound PGE$_2$ molecule (PDB:9R0M). **e** An interaction map of the residues within 10 Å of Zafirlukast from the three replicates of the simulation. **f** An interaction map of the residues within 10 Å of Losartan from the three replicates of the simulation. **g** Cell-based transport assays of human SLCO2A1 with Losartan ($n = 13$ independent experiments (mock); $n = 13$ (WT); $n = 7$ (R561A); $n = 13$ (W565A)) and Zafirlukast ($n = 12$ (mock); $n = 11$ (WT); $n = 6$ (R561A); $n = 12$ (W565A)); data shown are the mean and error bars are s.d.

Tolcapone are 1.12, 2.12, 45.2 and 24.1 μM, respectively (Fig. 3b). The values for Fentiazac and Tolcapone are considerably higher compared to Zafirlukast and Losartan, suggesting these drugs might interact differently with SLCO2A1, which may explain why these drugs are inhibitors rather than substrates.

To further understand how these drugs interact with the transporter, we determined the cryo-EM structure of rat Slco2a1 bound to Zafirlukast at 3.2 Å and Losartan at 3.1 Å (Table 1 & Supplementary Figs. 5-6). The drug-bound structures adopt similar outward-open conformations as the apo structure. The r.m.s.d. between Slco2a1-apo and each of the ligand-bound states is 0.38 Å (apo vs. Zafirlukast) and 0.182 Å (apo vs. Losartan). Zafirlukast adopts an L-shaped configuration in the binding pocket, occupying a position similar to that of the endogenous PG ligands (Fig. 3c). The most striking observation is the lack of polar interactions between the drug and the binding site, consistent with our observations for the two PG molecules. The sulfonyl group, which is located at one end of the drug molecule, is positioned equidistant between the E-ring oxygens and C-carboxylate of the PG molecules discussed above, and it sits within 4 Å of Arg561, with the 2-methylphenyl group, which is stabilized by Phe212 on TM 5. In the modelled position, the drug does not interact with Trp565; however, a 90° rotation of the carbamoyl moiety would bring the 2-methylphenyl group within ~ 3.0 Å of Trp565, which might easily occur following a conformational change in the transporter or the drug, and was observed in our MD simulations (Supplementary Fig. 7c). The methoxy group sits near the C15-hydroxyl group observed in PGE$_2$, whilst the remainder of the drug extends along the

hydrophobic surface of the binding site. Specifically, the drug packs against Val216 and Pro219 on TM5, Ile341 and Leu344 on TM 7 and Ile367, Asn371, Leu372 and Ala375 on TM 8, while the cyclopentyl ester, which is located at one end of the drug, packs against Tyr223 at the cytoplasmic end of the binding pocket. The phenyl and 3-methyl indole groups, which are present in the middle of the drug molecule, extend behind TM 5 and into the minor site, which is occupied by the α-chain of the PG molecules. Somewhat counterintuitively, therefore, the much larger molecule of Zafirlukast can mimic the key interactions observed for the endogenous PG ligands.

In contrast, Losartan adopts a C-shaped configuration in the binding site (Fig. 3d), with the biphenyl group oriented vertically relative to the membrane, positioning the tetrazole group near Trp565 (~ 4 Å) and occupying a position similar to the E-ring of the PG molecules. The remainder of the biphenyl group packs against Val216 and Pro219 in the N-terminal bundle, as well as the hydrophobic band near Ile341 in the C-terminal bundle. The 4-chloro-imidazol-5-methanol group occupies a position comparable to the C1-carboxylate observed in the PG molecules and sits near Phe212, Met379, His533, and Met560. The 2-butyl group extends into the C-terminal bundle, occupying a newly formed cryptic hydrophobic cavity created following a rotamer rearrangement in Phe337 in TM 7. Several similarities can be observed in how Losartan and Zafirlukast interact with Slco2a1 when compared to PGE$_2$ and PGF$_{2α}$ (Fig. 3c, f). The most notable aspects include interactions with Trp565, the presence of a negative charge near the C1-carboxylate binding position, adjacent to Arg561 and engagement with the prominent hydrophobic band. Indeed, mutation of Arg561 or

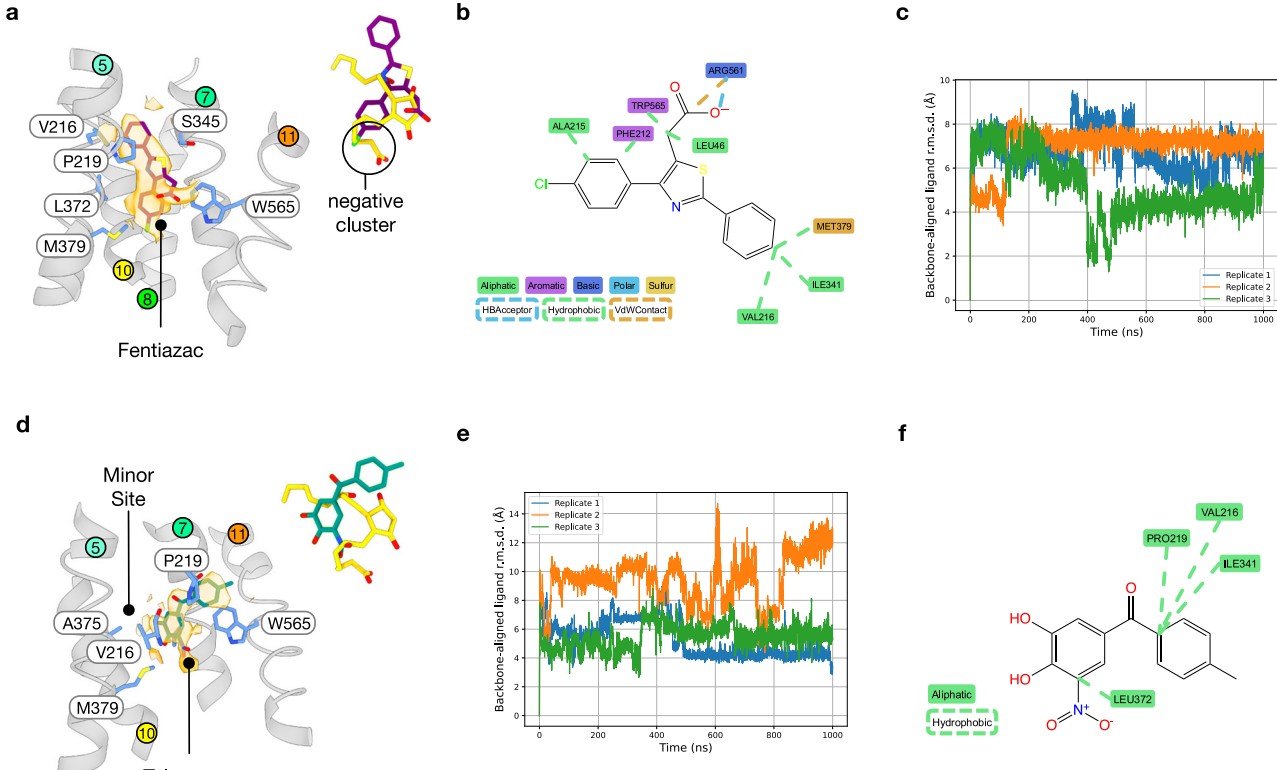

**Fig. 4 | Inhibitor binding in Slco2a1. a** A close-up view of the binding site reveals the position of Fentiazac. Cryo-EM volume density is depicted (orange and threshold 0.7). Inset–A structural overlay with the bound PGE2 molecule. **b** An interaction map of the residues within 10 Å of Fentiazac from the three replicates of the MD simulation. **c** 2D plot showing backbone-aligned ligand r.m.s.d values of the Fentiazac position in three replicates of the 1μs MD simulation. **d** A close-up view of the binding site reveals the position of Tolcapone. Cryo-EM volume density is depicted (orange and threshold 0.7). Inset–A structural overlay with the bound PGE2 molecule. **e** 2D plot showing backbone-aligned ligand r.m.s.d values of the Tolcapone position in three replicates of the 1μs MD simulation. **f** An interaction map of the residues within 10 Å of Tolcapone from the three replicates of the MD simulation. Only interactions present in more than 50 % of the simulation frames are shown.

Trp565 to alanine reduced the ability of SLCO2A1 to transport both drug molecules (Fig. 3g). The mutations had a greater effect on Zafirlukast uptake, possibly due to the more extensive interactions observed between this drug and the transporter.

To gain further insight into how these drugs interact with Slco2a1, we conducted three 1 μs simulations using the same methodology applied to the PG molecules discussed above (Supplementary Fig. 7). Clustering analysis of the MD trajectories reveals that both Zafirlukast and Losartan are relatively stable within the binding site, with ligand r.m.s.d values averaged over all trajectories of 4.30 Å ± 0.45 for Zafirlukast and 5.0 Å ± 0.99 for Losartan. However, these values compare favourably with the r.m.s.d values for PGE2 and PGF2α. Within the simulation frames, the top three clusters accounted for 99.2 % of the positions sampled by Zafirlukast, whereas for Losartan, this figure dropped to 75.9 %, aligning with the values obtained for PGF2α. Our analysis, therefore, indicates that Zafirlukast has fewer degrees of freedom within the binding site than Losartan, consistent with the higher-quality volume observed for this ligand.

Finally, we analyzed the interactions made by the drugs within the binding site (Fig. 3e, f). There is a notable difference between the two drugs: Zafirlukast makes extensive interactions with the binding site, whereas Losartan forms very few interactions with nearby aliphatic side chains. The most notable hydrogen bond interactions with Zafirlukast involved Gln42 and Ser345, whereas in Losartan, we observed hydrogen bonds only with Trp565 in 10% of the frames. Overall, the binding modes are dominated by hydrophobic, van der Waals and aromatic interactions, similar to our observations for the PG molecules.

## Structural basis for inhibitor recognition

Our results demonstrate that while Zafirlukast and Losartan are transported by human SLCO2A1, Fentiazac and Tolcapone are not under our assay conditions. Therefore, Fentiazac and Tolcapone are likely to act as inhibitors of PG transport (Fig. 3a). To gain insight into why these drugs are not transported, we determined the cryo-EM structure of rat Slco2a1 bound to either Fentiazac or Tolcapone at 3.1 Å (Supplementary Table 1 & Supplementary Figs. 8–9). As with the two structures bound to Zafirlukast and Losartan, the structures of Slco2a1 bound to Fentiazac and Tolcapone also adopt outward-open conformations. The r.m.s.d. between Slco2a1-PGE2 was 0.37 Å (PGE2 vs. Fentiazac) and 0.51 Å (PGE2 vs. Tolcapone). Fentiazac binds in the same pocket as the PG molecules, adopting a similar vertical orientation to Losartan (Fig. 4a). The pseudo-symmetrical structure of the compound made it challenging to model its orientation, and we leave open the possibility that it might also engage in the opposite orientation. However, using our prior observations from the PG-bound structures, we modelled the drug with the chloride atom occupying a position similar to that of the carboxylates, which we termed the negative cluster.

Nevertheless, in either orientation, the carboxylate group in Fentiazac occupies the same position as the E-ring in PGE2 and PGF2α. Although we don't observe a direct interaction between the carboxylate and Arg561 in the binding site, the cryo-EM volume around Arg561 extends towards the drug molecule (Supplementary Fig. 11a). When we analyzed the stability of the drug in the binding site using MD, we observed that Arg561 moves to form a salt bridge interaction with the carboxylate (Fig. 4b & Supplementary Fig. 11b-c) and interacts

with the drug with a frequency > 60 %. The remaining dominant interactions with the drug are van der Waals and hydrophobic, with a minor Pi-stacking interaction with Phe212, Trp565, and Phe557, which are similar to those observed in the PG-bound structures.

However, the r.m.s.d. values averaged over all trajectories are 6.29 ± 1.05 (Fig. 4c), which is notably higher than for either PG-molecule or the transported drugs, indicating the drug is not held as stably in the binding site. The phenyl-thiazole group extends outward toward the cytoplasmic side of the binding pocket and packs against Pro219, Val216, Ser345, and Leu372, as observed in the structures of Zafirlukast and Losartan. In this orientation, Fentiazac sits almost 90° relative to the position of the ω-chain in the PG structures and does not engage the minor pocket. Although the chlorophenyl group occupies the same position as the α-chain in PG-bound structures, the chloride atom does not occupy the same position as the C1-carboxylate. Instead, the chloride atom is located closer (~ 3.2 Å) to Met379 on TM 8 and (~ 4.0 Å) to His533 on TM10.

We also observed strong density in the binding site adjacent to Fentiazac (Supplementary Fig. 11d), which allowed us to model a Lauryl Maltoside Neopentyl Glycol (LMNG) detergent molecule into the structure. The LMNG molecule enters the binding site through the lateral opening noted in the PG-bound structures. One of the two maltopyranoside groups wedges between TM 5 and TM 8, whereas one of the didecylpropane chains extends down into the binding site, adjacent to the phenyl-thiazole group of Fentiazac and close to Leu348 and Ile367. The detergent density in Fentiazac partially overlaps with that of cholesterol hemisuccinate (CHS), which is observed in our PGE$_2$ and apo structures. As discussed below, the location of these lipophilic molecules within the lateral opening in the transporter suggests that ligands gain access to SLCO2A1 via the membrane.

In the Tolcapone bound structure, we again suffered from limited resolution and pseudo-symmetry in the molecule. Using the same rationale as we applied to the Fentiazac compound, we modelled the nitro group near the α-chain of the PG substrates, and the carbonyl group occupies the same position as the C15-hydroxyl group on the ω-chain of PGE$_2$ (Fig. 4d). In this orientation, tolcapone does not interact with Trp565 nor position a negatively charged group near the E-ring or C1-carboxylate binding sites in the transporter. We also observe that Arg561 adopts two rotamer positions in the cryo-EM volume (Supplementary Fig. 11e), indicating that the drug fails to engage Arg561 and that Arg561 is capable of adopting two rotamer positions within the transporter. Tolcapone also fails to interact with the hydrophobic band in this orientation. Instead, it binds near Pro219, Val216, Ala375, Met379, Ile341, and Ser345, similar to Fentiazac. However, Tolcapone extends several polar hydroxyl groups into the minor site, which may account for its orientation in the binding site. In the associated MD analysis of this binding pose, we observed considerable movement of the drug in the binding site (Fig. 4e & Supplementary Fig. 11f-g). The r.m.s.d values averaged over all trajectories are 6.73 ± 1.26, which is similar to that of Fentiazac and indicates this drug is also unable to adopt a stable binding pose. Or at least, a pose that is as stable as those observed for the PG molecules. The main interactions between the drug and the transporter are dominated by hydrophobic interactions made to Pro219, Val216, Ile341 and Leu372 (Fig. 4f & Supplementary Fig.11e). However, we did observe some pi-stacking interactions between the drug and Trp565 (Supplementary Fig. 11g), which may explain why this drug has some affinity for the binding site. Further free-energy calculations will be needed to follow up on these preliminary simulations.

In summary, the two non-transported drug molecules exhibit characteristics of substrate mimicry. Both Fentiazac and Tolcapone contain hydrophobic groups that interact with the same regions of the PG binding site that accommodate the aliphatic side chains, while their negatively charged polar groups fit into pockets that recognise the E-ring and the C1-carboxylate groups. However, unlike PGE$_2$, PGF$_{2\alpha}$,

Zafirlukast, and Losartan, neither Fentiazac nor Tolcapone appears capable of adopting a stable binding pose in the transporter and effectively engaging Arg561, which, as discussed below, we propose is essential for triggering transport across the membrane.

## Alternating access transport in SLCO2A1

In the MFS, alternating access transport is generally understood to occur through the concerted movement of four pairs of helices comprising the extracellular and intracellular gates, respectively[44,52]. In the outward-open state, as observed for rat Slco2a1 (Fig. 1a), the intracellular gate is formed by TM 4 and TM 5 packing against TM 10 and TM 11. The extracellular gate, formed by TM 1 and TM 2 and TM 7 and TM 8, is splayed apart in the open state. During the transition from outward-open to inward-open states, the helices of the extracellular gate must close to seal the binding site from the extracellular space. The closure of the extracellular gate, coupled with ligand binding in the central pocket, subsequently triggers the opening of the intracellular gate and the release of the ligand into the cytoplasm through a symporter mechanism[53].

To understand how this mechanism operates in human SLCO2A1, we compared our cryo-EM structure of PGE$_2$-bound rat Slco2a1 (PDB:9QZO) with the AlphaFold-predicted structure (Uniprot: Q00910)[54], which adopts an inward-facing state. The structural overlay reveals that the primary movement during transition from outward to inward-facing states occurs in the N-terminal bundle rather than the symmetrical movement of the N- and C-terminal bundles observed in some members of the MFS[28]. In our model, the N-terminal bundle rocks against the PGE$_2$ molecule, which is positioned horizontally via the interactions with Arg561, Trp565 and the hydrophobic band in the binding pocket (Fig. 5a). In transitioning from the outward-open to inward-open state, TM 1 and TM 2 rotate ~40° towards TM 7 and TM 8 and are stabilized in the closed state by two salt bridges between Glu60 and Lys61 on TM 1 with Lys350 (Lys351 in human SLCO2A1) and Glu353 (Glu354 in human SLCO2A1) on TM7 (Fig. 5b). While Lys61Ala and Glu354Ala variants had little impact on transport, substituting Glu60 or Lys351 with alanine abolished transport, indicating the importance of the Glu60-Lys351 salt bridge in the transport mechanism (Fig. 5c). On the cytoplasmic side of the transporter, the intracellular gate helices, TMs 4 and 5, move away from TMs 10 and 11 along an axis originating from Pro219, executing a ~ 28° rotation and breaking the salt bridge between Asp196 and Asp197 on TM4 and Arg542 and Lys549 on TMs 10 and 11, respectively. Substituting Asp196 or Asp197 with alanine abolished transport, while Arg542Ala retained approximately 50% activity (Fig. 5c). A similar set of salt bridge interactions was noted in the SLCO1B1 structure[39], suggesting these are conserved features of the SLCO superfamily.

An important question for all transport mechanisms is how ligand binding is coupled to the conformational changes that lead to substrate transport. To understand the structural implications for substrate binding, we determined the ligand-free structure of rat Slco2a1 to 2.9 Å (Supplementary Table 1 and Supplementary Fig. 10). As we observed in the previous ligand-bound structures, Slco2a1 adopts the same outward-facing state, with an r.m.s.d of 0.32 Å vs. the PGE$_2$-bound structure. An overlay of the structures obtained in the presence of either PGE$_2$ or PGF$_{2\alpha}$, compared to the Apo structure, illustrates that, within the binding pocket, the only significant movement following PG binding occurs in the rotamer position of Arg561 (Fig. 5d). In the Apo structure, Arg561 on TM11 forms a salt bridge interaction with Glu78 on TM2. In the PGE$_2$-bound state, Arg561 and Glu78 change their rotamers but remain close enough to maintain the salt bridge. However, in the PGF$_{2\alpha}$ structure, Arg561 points down towards the C1-carboxylate of the substrate, while Glu78 points away towards the extracellular gate and interacts with Lys53

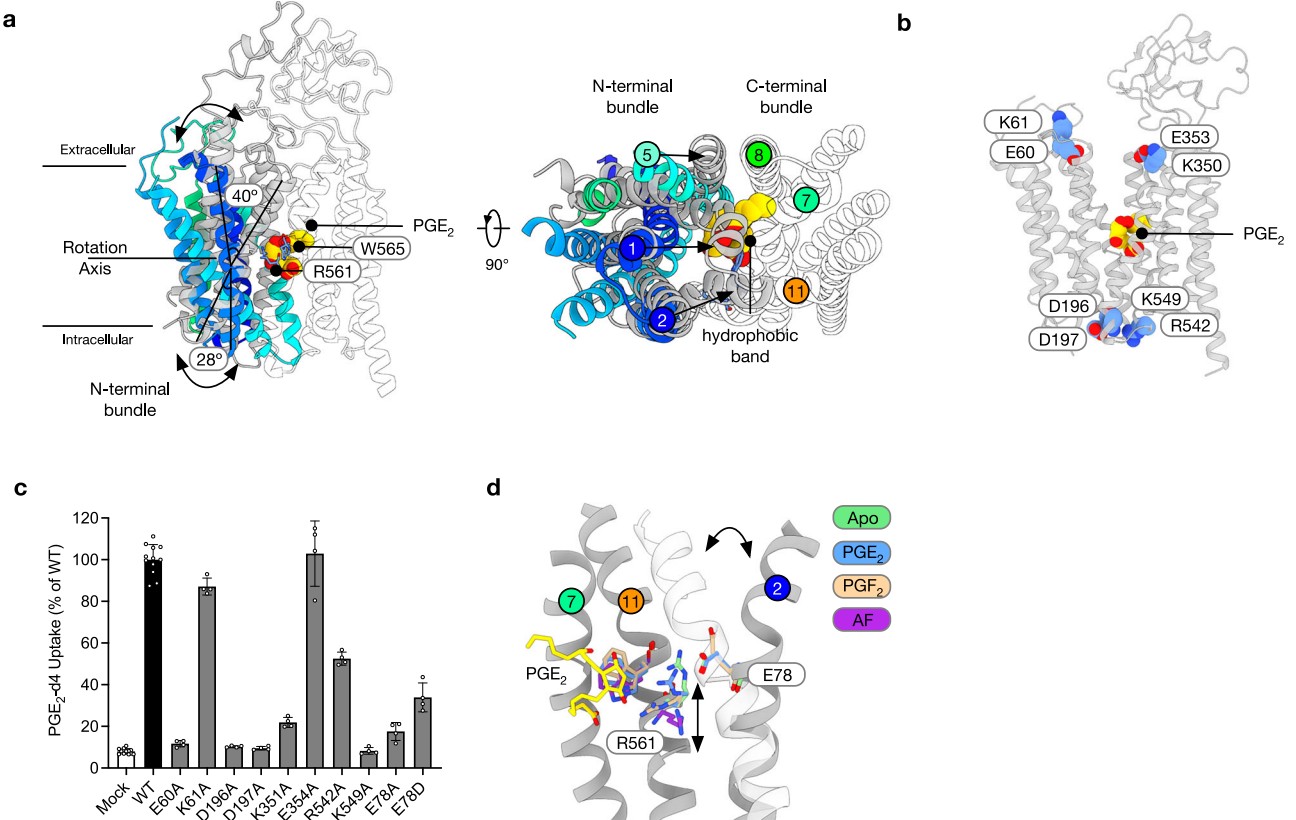

**Fig. 5 | Conserved salt bridge interactions coordinate PG binding and transport. a** Structural overlay of the PGE2-bound Slco2a1 with the inward-facing AlphaFold2 model (AF-Q00910-F1). The main structural differences are evident in the N-terminal bundle, which rotates around the bound PGE2 substrate, represented by spheres. The axis of rotation traverses the hydrophobic band formed and runs through Trp565 and Arg561, resulting in TMs 1, 2, and 5 packing against TMs 7, 8, and 11 in the inward-facing state. **b** Conserved pairs of salt bridge interactions coordinate the arrangement of the extracellular and intracellular gate helices, located above and below the central PG binding site. **c** Cell-based transport assays for wild type (WT) and mutant human SLCO2A1; $n = 12$ independent experiments for the WT and the mock-transfected control; $n = 4$ for the mutants; data shown are the mean and error bars are s.d. **d** A zoomed-in view of the binding site illustrating the structural overlay between the Apo, PG-bound, and AF models of Slco2a1. The direction of movement observed in Arg561 and TM5 and TM2 are indicated.

on TM1, breaking their interaction. The density around Glu78, however, is weak, suggesting that Arg561 could move away without Glu78 needing to adopt the alternate rotamer. Additionally, the dual rotamer position for Arg561 in the Tolcapone-bound structure confirms the ability of Arg561 to change rotamer positions in response to ligand binding (Supplementary Fig. 11e). For Slco2a1 to transition to the inward-facing state, the 40° rotation on TMs 1 and 2 brings Glu78 into the position formally occupied by Arg561 in the outward open Apo state. At the same time, Arg561 moves further down into the pocket, which may help pull the PG towards the cytoplasmic exit tunnel, which opens as TMs 4 and 5 move away from TMs 10 and 11.

Arginine 561 and Glu78 are strictly conserved within the SLCO superfamily and are essential for PG transport[9,49,55], with even a conservative substitution of Glu78 to aspartate reducing function by > 60 % (Fig. 5c). The equivalent arginine in SLCO1B1 and SLCO1B3, Arg580 (Supplementary Fig. 1), is also essential for function[56,57], similar to our observation in SLCO2A1 (Fig. 1d). Our model therefore suggests that ligand recognition and transport may be separate events within the OATP family. Each member of the OATP family has specific side chains that recognise their respective ligands; in the case of SLCO2A1, this would be principally Trp565. Once successfully bound, the ligands would then disrupt the Arg-Glu interaction, triggering transport. This mechanism is an elegant way for the SLCO superfamily to evolve specific ligand recognition mechanisms whilst retaining a conserved mechanism to trigger alternating access transport. The proposed alternating access mechanism may also explain why Fentiazac and Tolcapone are not substrates, as these

drugs are unable to adopt a stable binding pose within the transporter (Fig. 4c–e), restricting their ability to disrupt or otherwise modulate the salt bridge between Glu78 and Arg561.

### Lipid densities suggest ligand entry via the membrane

Prostaglandins have high logP values and are highly soluble in lipid environments, prompting us to question how these molecules access the binding site from the extracellular space. As noted above, the two six-helix bundles in SLCO2A1 splay apart sufficiently to open the binding site to the lipid bilayer via two lateral openings (Fig. 1a). In our cryo-EM maps for the apo and ligand-bound structures, we observe a concentration of lipid-like densities clustering within this opening, specifically between TM5 from the N-terminal bundle and TM8 from the C-terminal bundle. The position of these lipids is in a very similar position to the density for LMNG observed in the Fentiazac complex (Supplementary Fig. 11d), although they do not fully enter the binding site. In the PG-bound and apo structures, we modelled this density as CHS due to the planar profile (Fig. 6a). Similar lipid-like densities were observed in the organic cation transporter OCT3 (SLC22A3)[58] and SLCO1B1[39].

The location of the CHS density, which sits above the ω-chain of PGE2 and PGF2α, prompted us to speculate that PG molecules may enter the transporter from the membrane. Two mutations in SLCO2A1, which are located near the CHS molecule on TM5 (Gly222Arg) and TM8 (Gly369Asp), were previously identified in patients diagnosed with PHO (Fig. 6b). These mutations result in a severe reduction of activity (Fig. 6c). The CHS also sits near Tyr223 and Ala220 on TM5.

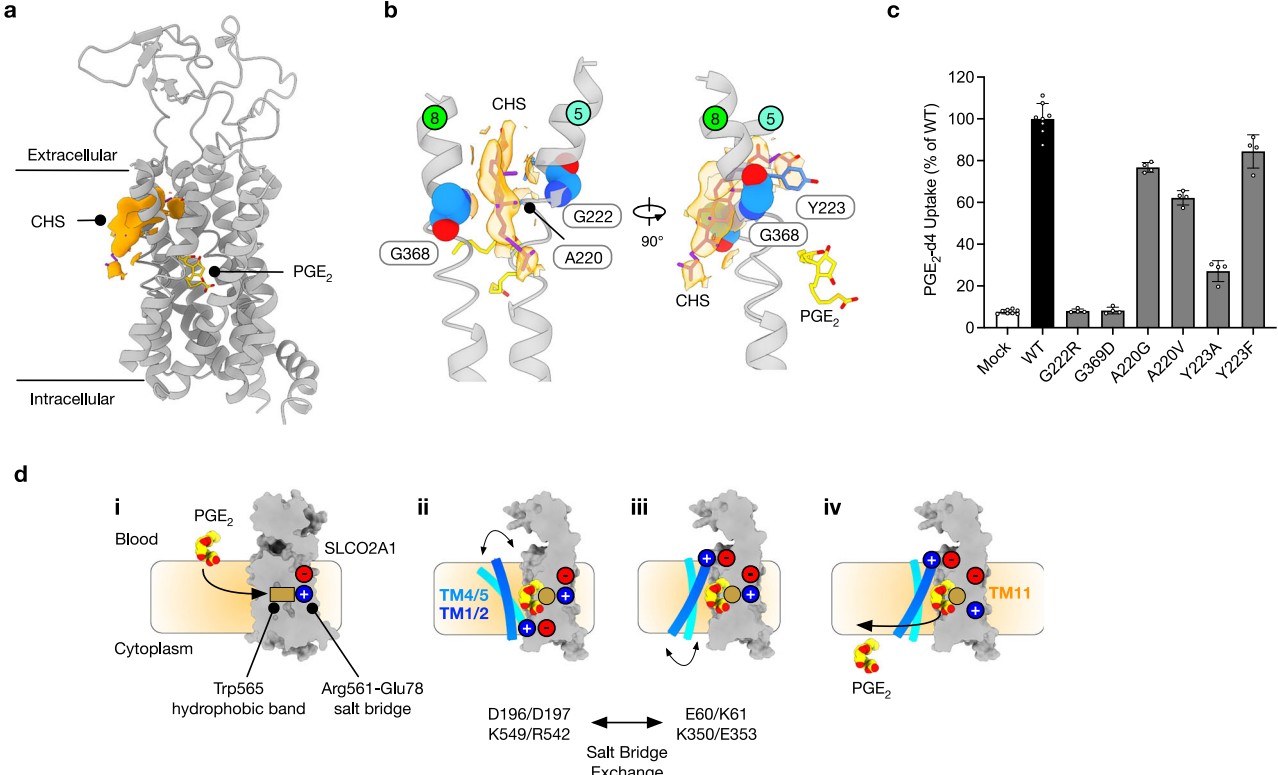

**Fig. 6 | Alternating access transport in SLCO2A1. a** A CHS molecule can be modelled into the cryo-EM volume density observed in the vestibule that opens into the membrane between TMs 5 and 8. Cryo-EM volume density is shown (orange and threshold 0.2). **b** A zoomed-in view of the lateral opening and the cryo-EM volume density (orange and threshold 0.2). Residues within 5 Å of the bound CHS are shown. Note, Gly368 in rat SLCO2A1 is equivalent to Gly369 in human SLCO2A1. **c** Cell-based transport assays for wild type (WT) and mutant human SLCO2A1. $n = 8$ independent experiments for the WT and the mock-transfected control, and $n = 4$ for the mutants; data shown are the mean and error bars are s.d. **d** Schematic model of the transport mechanism in Slco2a1, highlighting key structural features and their role in ligand binding and conformational changes in the transporter. (i) opening observed between TM 5 and TM 8 in the outward-open state (ii) transition to the inward-open state via the movement of TM 1 and TM 2, (iii) AF2 model of the inward-open state, (iv) PG exit from the binding pocket into the inner leaflet of the membrane.

Substituting Tyr223 with Ala also reduced activity by approximately 85 %, while a conservative phenylalanine replacement retained activity, indicating the importance of a hydrophobic group at this position. Similarly, substituting Ala220 with valine reduced activity by approximately 40 %, whereas a glycine replacement retained activity, indicating that maintaining the space between TMs 5 and 8 is also essential. Our results, combined with the location of the PHO disease mutations, suggest that this region of SLCO2A1 is crucial for function and support the hypothesis that some PG and drug molecules may enter the transporter from the membrane.

### Prostaglandin and drug transport via SLCO2A1

Taken together, our structural, functional and modelling data suggest a mechanism for PG and drug transport via SLCO2A1 (Fig. 6d). In the outward-open state, ligands access the binding pocket from the outer leaflet of the plasma membrane, likely through the opening observed between TM 5 and TM 8 (i). The presence of several basic side chains in combination with the hydrophobic band favours the recognition of negatively charged, lipophilic molecules, such as PGs, and the chemically similar drug molecules discussed above. The interaction with Trp565, along with the conserved hydrophobic band, positions the PG in a productive binding pose. In this position, the E-ring aligns with the rotation axis necessary to transition to the inward-open state via the movement of TM 1 and TM 2 to close the extracellular gate (ii). The E-ring and C1-carboxylate groups disrupt the salt bridge between Arg561 and Glu78, weakening the interaction between the extracellular gate helices TM 5 and TM 8. The specific role of the Arg561-Glu78 salt

bridge in the mechanism warrants further investigation, as the conformational change between outward and inward-facing states also involves the pairs of salt bridges in the extracellular and intracellular gates mentioned above. However, Arg561 and Glu78 are strictly conserved within the SLCO superfamily, suggesting this interaction may form part of a conserved mechanism.

Comparison with the inward-open AF2 model indicates that when the transporter transitions to the inward-open state (iii), TM 11 rotates inwards, pulling Arg561 away from the C1-carboxylate and restoring its interaction with Glu78. Reforming the Arg561-Glu78 bond would eliminate the favourable interaction with the C1-carboxylate and could be described as a substrate hand-off model, with Arg561 serving as the trigger for the conformational change. Finally, in the inward-open state, a similar cavity opens into the inner leaflet of the membrane between TMs 5 and 8, directly adjacent to the C1-carboxylate. The PG could then exit the binding pocket into the inner leaflet of the membrane, and subsequently into the cytosol (iv). An interesting aspect of our mechanism is the potential role of the membrane as a selectivity filter. Only sufficiently lipophilic molecules would gain access to the transporter binding site, wherein only those that can orient correctly and stably to engage Arg561 trigger the transport cycle.

## Discussion

The SLCO/OATP family is essential for regulating the flow of metabolites, drugs and xenobiotics throughout the body[59,60]. Members of the SLCO1 family, such as SLCO1A2 (OATP1A2), SLCO1B1 (OATP1B1) and SLCO1B3 (OATP1B3), are generally polyspecific and transport a broad

range of amphipathic organic compounds[8]. The SLCO2 subfamily, however, has a narrower range of ligand selectivity, predominantly restricted to PGs and thromboxanes[22,45,47]. How, then, do different members of the SLCO family select between different families of metabolites and drugs? To partly address this question, we can compare recent structures of SLCO1B1 in complex with several different endogenous and exogenous ligands[38,39] with the structures presented in this study for SLCO2A1. A key finding from recent structural studies of SLCO1B1 and SLCO1B3 is that ligand binding is predominantly driven by hydrophobic interactions, in line with our observations here for SLCO2A1. Specifically, the binding pocket in SLCO1B1 contains a major and a minor binding site[38,39]. The major binding pocket is similar to that observed in SLCO2A1 and is lined with aromatic and hydrophobic side chains, which can contribute to van der Waals interactions with the ligands. The $PGE_2$ ligand overlays with the structures of SLCO1B1 bound to bilirubin, an endogenous ligand for this transporter (Supplementary Fig. 12). The interaction of SLCO1B1 with bilirubin, estrone-1-sulfate and several drug molecules is driven by interactions with three aromatic side chains on TM 7 and TM 8, Phe356, Tyr352 and Phe386[39]. These are absent in SLCO2A1 and replaced with Ser345, Ile341 and Ala375, respectively. In SLCO2A1, the aromatic interactions have moved to the opposite side of the major binding site, which is replaced with Gly584 and Ser576 in SLCO1B1. The minor site, where estrone-3-sulfate binds in SLCO1B1, extends from the major site into the C-terminal bundle[38,39].

In SLCO2A1, we observed that the α-chain of $PGE_2$ and $PGF_{2\alpha}$ interacted with the minor site, along with the cyclopentyl indole group of Zafirlukast. However, the cavity in SLCO2A1 is noticeably smaller than that in SLCO1B1 due to the presence of Ile341 and Leu344 on TM 7, which are required to interact with the PG substrates. These observations demonstrate that while hydrophobic and aromatic interactions drive substrate recognition within the SLCO superfamily, these are distributed differently within the binding pocket to orient specific sets of metabolites correctly. We suggest it is the correct orientation of the anionic group within all SLCO substrates with respect to the conserved Glu78 - Arg561 salt bridge that serves to functionally couple ligand binding to alternating access transport within the broader SLCO/OATP superfamily.

Finally, given the promiscuous nature of the binding sites in the SLCO/OATP transporter superfamily, one way for the cell to control their function is to limit access to these sites to specific sets of metabolites. In SLCO2A1 and other family members, access appears to be controlled by the wide vestibule that opens into the membrane, which may represent an essential component of the regulatory mechanism governing PG transport.

## Methods

### Cloning, expression, and purification of Slco2a1

The gene encoding full-length (residues 1-644) *Rattus norvegicus* Slco2a1 (UniProt: Q00910) was synthesized as a DNA fragment and cloned into pLexM vector[61] containing a C-terminal Avi-tag upstream of His-tagged green fluorescent protein (GFP) and a TEV cleavage site between Avi-tag and the GFP. The plasmid was transiently transfected with PEI-MAX (Polyscience Cat. No. 24765-1), into human embryonic kidney-293 F cells as previously described[62]. HEK293F cells were cultured as a suspension in FreeStyle™ 293 Expression Medium (Life Technologies Cat. no. 12338026) at 37 °C and 8 % $CO_2$. About 18–24 h prior to transfection, cells were seeded at a density of $0.7 \times 10^6$ cells/mL to achieve an optimal density of $1.3–1.4 \times 10^6$ cells/mL. For transfection, 2.2 mg of plasmid DNA and 4.5 ml of PEI MAX (1 mg/ml stock) were diluted in Dulbecco's Modified Eagle Medium (Thermo Fisher Scientific), combined, and incubated at room temperature for 10–15 min to allow complex formation. The resulting DNA-PEI complex was added dropwise to the cells with gentle swirling, followed by the addition of 8–9 mM sodium butyrate to enhance

transgene expression. Cells were harvested 42 h post-transfection, resuspended in 1 x PBS and frozen until required. Membranes were prepared by lysing the cells via sonication, and unbroken cells and cell debris were pelleted at 12,000 x *g* for 10 min at 4 °C. Membranes were harvested through centrifugation at $200,000 \times g$ for 1 h and washed once with 15 mM HEPES, pH 7.5, 20 mM KCl. After washing, the membranes were resuspended in PBS and snap-frozen at −80 °C for storage until required.

For purification thawed membranes (~7–8 g wet weight) were solubilized in 130 mL of buffer containing 1 x PBS supplemented with an additional 150 mM NaCl and 10% glycerol, 1% (wt/v) Lauryl Maltoside Neopentyl Glycol (LMNG) detergent (Anatrace Cat. No. NG318) and 0.1% (wt/v) Cholesterol hemisuccinate (CHS; Merck Cat. No. C6512) for 2 hours under gentle agitation using a magnetic stir plate. All the purification steps were carried out at 4 °C. Insoluble material was removed through centrifugation for one hour at $200,000 \times g$, and 15 mM imidazole was added to the supernatant. Slco2a1 was purified to homogeneity using standard immobilized metal-affinity chromatography. For batch resin binding, 4 mL of HisPur-Ni-NTA resin (Fisher Scientific Cat. No. 10038124) was incubated for three hours with gentle stirring. The resin was loaded onto a gravity flow column (BioRad) and sequentially washed with 10 CVs of buffer with 15 mM imidazole and 0.1% LMNG:CHS (10:1), 15 CVs of buffer with 0.05% LMNG:CHS (10:1), 15 mM imidazole, 1 mM ATP (pH 7.0), and 2 mM $MgCl_2$, followed by 15 CVs of 20 mM imidazole buffer to remove ATP. The protein was eluted in five CVs of buffer containing 250 mM imidazole and dialyzed overnight with TEV protease (1:0.5 M ratio) against 20 mM Tris pH 7.5, 150 mM NaCl with 0.003 % LMNG: CHS (10:1) at 4 °C. Following TEV cleavage, the protease and cleaved His tagged GFP were removed through nickel affinity chromatography as described above, after adding 10 mM imidazole to the dialyzed sample. The protein was then concentrated to 500 µL using a Vivaspin-20 30 KDa MWCO spin concentrator (Sartorius) at 4 °C and subjected to size exclusion chromatography (Superdex 200 increase 10/300 GL column, VWR Cat. No. 28-9909-44) in a buffer consisting of 25 mM Tris pH 7.5, 150 mM NaCl with 0.0015 % LMNG and 0.00015 % CHS. The monodisperse peak fractions containing Slco2a1 protein were pooled, concentrated to 2–3 mg.ml$^{-1}$ using Vivaspin-500 (30KDa MWCO, Sartorius), snap frozen and stored at −80 °C in small aliquots.

### Cryo-EM sample preparation and data acquisition

For substrate and drug bound complexes, the purified Slco2a1 protein (2 mg/ml) was pre-incubated with the different compounds at final concentrations of 1.4 mM Prostaglandin $E_2$ (Cayman Chemical Cat. No. 14010), 0.5 mM for Tolcapone (Sigma Cat. No. SML0150), Fentiazac (MedChemExpress Cat. No. HY-118752) and Losartan Potassium (Sigma Cat. No. SML3269), 0.25 mM for Prostaglandin $F_{2\alpha}$ (European Pharmacopoeia D2255000) and Zafirlukast (Sigma Cat. No. Z4152), on ice for at least 30 minutes prior to grid preparation. Except for losartan potassium, which was water soluble, the stock solution for compounds was made in DMSO (Sigma Cat. No D8418). Four µL of protein sample with compounds or without (for apo structure), were adsorbed to glow-discharged holey carbon-coated grids (Quantifoil 300 mesh, Au R1.2/1.3) for 7–10 s. Grids were then blotted for 2 or 4 s with blot force ranging from +5 to −25 at 100 % humidity between 4–10 °C and frozen in liquid ethane using a Vitrobot Mark IV (Thermo Fisher Scientific). Movies for the Apo, $PGE_2$, $PGF_{2\alpha}$ and Tolcapone datasets, movies were collected in counted mode, in Electron Event Representation (EER) format, on a CFEG-equipped Titan Krios G4 (Thermo Fisher Scientific) operating at 300 kV. A Selectris X imaging filter (Thermo Fisher Scientific) was used, with slit width of 10 eV, along with a Falcon 4i direct detection camera (Thermo Fisher Scientific) corresponding to a calibrated pixel size of 0.732 Å at 165,000x magnification. Movies were collected with a defocus range of −2.0 to −0.5 µm at a total dose of 55.0 to 61.2 e-/Å$^2$ (Supplementary Table 1), fractionated to ~1.0 e-/Å$^2$ per

fraction for motion correction. Fentiazac and Zafirlukast datasets were collected in counted super resolution mode on a Titan Krios G3 (FEI) with a K3 camera (Gatan) and BioQuantum imaging filter at 300 kV, with a pixel size of 0.832 Å. Movies were collected with a defocus range of −2.4 to −1.0 μm at a total dose of 40.2–49.4 $e^-/Å^2$ fractionated to ~1 $e^-/Å^2$ per frame.

## Cryo-EM data processing

Initial micrograph processing was performed in SIMPLE 3.0[63] using SIMPLE-unblur for patched (15 ×10) motion correction, SIMPLE-CTFFIND for patched contrast transfer function estimation and SIMPLE-picker for particle picking (300 × 300 box size) and particle extraction. All downstream processing was done in cryoSPARC 4.5.3[64] or RELION 4.0[65], using the csparc2star.py script within UCSF pyem[66] to convert between formats. Global resolution was estimated from gold-standard Fourier shell correlations (FSCs) using the 0.143 criterion, and local resolution estimation was calculated within cryoSPARC.

The cryo-EM processing workflow for Slco2a1 with bound $PGE_2$ is outlined in Supplementary Fig. 2. Briefly, particles were subjected to two rounds of reference-free 2D classification ($k = 200$ each) using a 140 Å soft circular mask within cryoSPARC. Selected particles (1,405,443) were subjected to multi-class ab initio reconstruction using a maximum resolution cutoff of 5 Å, generating four volumes. These volumes were lowpass-filtered to 8 Å and used as references for a heterogeneous refinement against the same particle subset. Particles (487,353) from the most populated class were selected and non-uniform refined against their corresponding volume, lowpass-filtered to 15 Å, generating a 3.7 Å map. Further pruning was performed by subjecting the particle subset to multi-class ab initio reconstructions and selecting particles (429,600) from three of the four generated classes, demonstrating strong transmembrane helix density. Non-uniform refinement of these particles against a previous volume, lowpass-filtered to 15 Å, yielded a 3.6 Å reconstruction, which was further improved to 3.4 Å by local refinement using a soft mask encompassing the protein component of the volume. Bayesian polishing in RELION, followed by 2D classification and non-uniform refinement in cryoSPARC (plus fitting per-particle defocus parameters) yielded a 3.2 Å volume from 327,674 particles.

The cryo-EM processing workflow for Slco2a1 with bound $PGF_{2\alpha}$ is outlined in Supplementary Fig. 3. Briefly, particles were subjected to three rounds of reference-free 2D classification ($k = 200$ each) using a 140 Å soft circular mask within cryoSPARC. A subset of particles (1,065,537) was subjected to multi-class ab initio reconstruction using a maximum resolution cutoff of 6 Å, generating four volumes. These volumes were lowpass-filtered to 8 Å and used as references for a heterogeneous refinement against the full 2D-cleaned particle set, yielding one class with clearly defined transmembrane helices. Particles (401,078) from this class were non-uniform refined against their corresponding volume, lowpass-filtered to 15 Å, generating a 3.7 Å map. These particles underwent an additional round of heterogeneous refinement against the same ab initio volumes, lowpass-filtered to 15 Å, resulting in a 3.8 Å map from 206,784 particles. Local refinement using a soft mask encompassing the protein component of the volume further improved map quality to 3.6 Å. Bayesian polishing in RELION followed non-uniform refinement in cryoSPARC (plus fitting per-particle defocus parameters) yielded a 3.4 Å map.

The cryo-EM processing workflow for Slco2a1 with bound Zafirlukast is outlined in Supplementary Fig. 5. Briefly, 15,947 movies were collected in four datasets. The extracted particles were combined and subjected to two rounds of 2D classification using a 150 Å soft circular mask. Four volumes were generated by multi-class ab initio reconstruction from a selected particle subset (3,321,479). The volume with visible TMS was refined by another round of ab initio reconstruction with the maximum resolution cutoff of 4 Å and subsequently used as a reference along with three other junk volumes for a 4-class

heterogeneous refinement (8 Å initial low-pass) against the full 2D-cleaned 4,823,165 particles set. Particles (1,888,803) from the most structured class were further sorted by a multi class ab initio reconstruction and the volume (887,765 particles) with most resolved TMs and extracellular domain features was non-uniform refined (15 Å low-pass filter) against its corresponding volume. Bayesian polishing (RELION 3.1) followed by an additional round of 2D classification ($k = 100$) resulted in a selection of 682,931 particles, which were non-uniformly refined, followed by CTF refinement and subjected to 3D classification ($k = 4$, filter resolution of 3.5 Å) with a soft protein mask. Particles (177,884) belonging to a class with strong ligand density were subjected to rebalance orientation (percentile 70) to overcome the anisotropy in the map. This process generated a 3.1 Å volume with cFAR score: 0.45 (146,876 particles), which was used for model building.

The cryo-EM processing workflow for Slco2a1 with bound Losartan is outlined in Supplementary Fig. 6. Briefly, 26,332 movies were collected in three datasets. The extracted particles were combined and subjected to two rounds of 2D classification using a 150 Å soft circular mask. The cleaned 2D classes were sorted into two sets based on orientation: Set 1 with 2D classes for side views of particles and Set 2 with top and bottom view classes. To overcome the issue of preferential orientation, set 2 particles (overpopulated views) were subjected to rebalance 2D classification with 10 super classes. Four volumes were generated by multi-class ab initio reconstruction with set 1 particles and rebalanced set 2 particles. The volume with visible TMs was refined by another round of ab initio reconstruction with the maximum resolution cutoff of 4 Å and subsequently used as a reference along with three other junk volumes for a 4-class heterogeneous refinement (8 Å initial low-pass) against the full 2D cleaned unbalanced particle (8,766,437) set. Particles were further sorted by 2D classification and another round of heterogeneous refinement. Particles (328,467) from the most structured class were non-uniform refined (15 Å lowpass filter) against their corresponding volume and produced a map with low cFAR score of 0.13. Bayesian polishing (RELION 3.1) followed by an additional round of 2D classification ($k = 100$) resulted in a selection of 294,031 particles which were subjected to CTF refinement and 3D classification ($k = 4$, filter resolution of 3.5 Å) with a soft protein mask. This resulted in a class (80,622 particles) with strong ligand density. Following non-uniform refinement against a 15 Å lowpass-filtered reference generated a 3.1 Å volume (cFAR score: 0.47) with good local resolution to be used for model building.

The cryo-EM processing workflow for Slco2a1 with bound Fentiazac is outlined in Supplementary Fig. 7. Briefly, 12,155 movies were collected. The extracted particles (6,788,854) were subjected to two rounds of 2D classification using a 150 Å soft circular mask. Four volumes were generated by multi-class ab initio reconstruction from a selected particle subset (2,768,399). The volume (839,055 particles) with visible TMs was refined by another round of ab initio reconstruction with the maximum resolution cutoff of 4 Å and subsequently used as a reference along with three other junk volumes for a 4-class heterogeneous refinement (8 Å initial low-pass) against the full 2D-cleaned 3,199,523 particles set. Particles (1,013,200) from the most structured class were further cleaned by 2D classification ($k = 100$), multi class ab initio reconstructions and another round of heterogeneous refinement. The particles (153,553) from the volume with most resolved TMS and extracellular domain features were non-uniformly refined (15 Å lowpass filter) against their corresponding volume, generating a 3.1 Å map with 0.25 cfar score. Bayesian polishing (Relion 3.1) and an additional round of 2D classification ($k = 80$) resulted in a selection of 144, 011 pruned particles. These particles underwent non-uniform refinement followed by CTF refinement (per-particle defocus refinement and per-group CTF refinement fitting beam tilt and trefoil) and were subjected to 3D classification ($k = 3$, filter resolution of 3.5 Å) with a soft protein mask. Particles (54,064)

belonging to a class with strong ligand density were non-uniformly refined against a 15 Å low-pass-filtered reference. This generated a 3.1 Å volume used for model building with an improved cFAR score of 0.45, alleviating the issue of preferential orientation.

The cryo-EM processing workflow for Slco2a1 with bound Tolcapone is outlined in Supplementary Fig. 8. Briefly, particles were subjected to two rounds of reference-free 2D classification ($k = 200$ each) using a 140 Å soft circular mask within cryoSPARC. Selected particles (2,776,900) underwent heterogeneous refinement against the four ab initio volumes generated from the dinoprost dataset, lowpass-filtered to 8 Å. Particles (808,151) from the most populated class were selected and non-uniform refined against their corresponding volume lowpass-filtered to 15 Å, generating an anisotropic 3.3 Å map that was improved by an additional round of heterogeneous refinement using the same reference volumes, followed by non-uniform refinement. Particles (415,336) from this reconstruction further underwent 3D classification in cryoSPARC using a resolution filter of 4 Å and a soft mask surrounding the protein component of the volume. Particles (131,095) belonging to the most populated class and structured volume underwent non-uniform refinement against their corresponding volume, lowpass-filtered to 15 Å, generating a much more isotropic reconstruction that could be further improved to a resolution of 3.3 Å by local refinement of the protein component using a soft mask. Bayesian polishing in RELION followed by non-uniform refinement in cryoSPARC (plus fitting per-particle defocus parameters) yielded a 3.1 Å map.

The cryo-EM processing workflow for Apo Slco2a1 is outlined in Supplementary Fig. 9. Briefly, particles were subjected to two rounds of reference-free 2D classification ($k = 200$ each) using a 140 Å soft circular mask within cryoSPARC. Selected particles (1,342,191) were subjected to multi-class ab initio reconstruction using a maximum resolution cutoff of 6 Å, generating four volumes. Particles (468,312) from the most populated class were selected for non-uniform refinement against their corresponding volume, and lowpass-filtered to 15 Å to generate the final 3.0 Å volume. Bayesian polishing in RELION, followed by non-uniform refinement in cryoSPARC, further improved map quality. Heterogeneous refinement of these polished particles against the previously generated four ab initio volumes, lowpass-filtered to 20 Å, improved volume quality to 2.9 Å after non-uniform refinement of the most populated class (220,549 particles).

## Model building and refinement

The AlphaFold 2 model of rat Slco2a1 was initially docked into the globally-sharpened map for the Apo and adjusted where necessary by manual building using Coot v. 0.9[67], ISOLDE[68] and real-space refinement in PHENIX v. 1.21.1-5286[69] using secondary structure, rotamer and Ramachandran restraints. Ligand restraints were generated using Grade2[70]. The final models were validated using MolProbity[71] within PHENIX. Figures were prepared using UCSF ChimeraX v.1.8[72].

## Site-directed mutagenesis of rat and human SLCO2A1

To determine the PG transport mediated by mutants of human SLCO2A1 and rat Slco2a1 genes inserted in pIRES2-EGFP, the indicated amino acid residues were replaced using the site-directed mutagenesis system with PrimeSTAR® DNA polymerase (Takara Bio Inc., Kusatsu, Japan) and the oligonucleotides with mutations of interest (see Source Data File). All oligonucleotides were synthesized by Integrated DNA Technology (Coralville, IW). Amplified PCR products were digested with Dpn I (Nippon Gene, Tokyo, Japan) and used to transform E. coli. Plasmids were prepared, and the mutated sequences were confirmed by Fasmac Co., Ltd. (Atsugi, Japan).

## Cell-based transport assay

Cell-based transport assays were performed as described previously[43,73]. For the assay of SLCO2A1 mutants, HEK293 cells

(RCB1637, Riken BioResource Research Centre, Tsukuba, Japan) were plated on a poly-L-lysine-coated four-well plate (Bio Medical Science Inc., Tokyo, Japan) at a density of $1.5 \times 10^5$ cells per well in antibiotics-free Dulbecco's modified Eagle's medium (DMEM, Nacalai Tesque Inc., Kyoto, Japan) supplemented with 10 % (v/v) foetal bovine serum (CCP-FBS-BR-500, Cosmo Bio Co., Ltd., Tokyo, Japan). The next day, cells were transfected with Lipofectamine 3000 reagent (Invitrogen Corp., 0.4 μg DNA per well) using human or rat SLCO2A1 constructs in the vector pires2-EGFP or vector alone (Mock) for 48 h. HEK293 cells stably expressing human SLCO2A1, which were previously prepared[23], were used for the drug transport and inhibition assay. For this assay, cells were washed once with transport buffer (125 mM NaCl, 4.8 mM KCl, 5.6 mM D-glucose, 1.2 mM CaCl$_2$, 1.2 mM KH$_2$PO$_4$, 1.2 mM MgSO$_4$, and 25 mM HEPES for pH 7.4) and then incubated with the buffer containing PGE$_2$-d4 (100 nM) or PGF$_{2\alpha}$ (100 nM) for 30 seconds in the presence or absence of the indicated inhibitor drug. Similarly, cellular uptake of Fentiazac (1 nM), Tolcapone (100 nM), Losartan (100 nM), or Zafirlukast (100 nM) was determined. At the end of the uptake, cells were washed with ice-cold buffer, collected, and lysed with a 70% acetonitrile solution containing 10 nM PGE$_2$-d9 as an internal standard. The supernatant of the lysed solution was dried under vacuum at 40 °C, and the residue was reconstituted with 0.1% formic acid/acetonitrile (2:1, v/v) for quantification with an LC-MS/MS system consisting of an LCMS 8040 triple quadruple mass spectrometer (Shimadzu Corp., Kyoto, Japan) coupled with an LC-20ADXR ultra-fast liquid chromatography system (Shimadzu).

The flow rate of the mobile phase was 0.4 mL/min, and the injection volume was set at 30 μl. CAPCELL PAK IF2 (C18, 2.1 mm I.D. x 100 mm, 2 μm, Osaka Soda Co. Ltd., Osaka, Japan) served as the analytical column at 40°c. Samples were maintained at 4 °C during the analysis. Electrospray ionization was employed, and the mass transitions were monitored at m/z 355.4/275.3 for PGE$_2$-d4, 360.3/280.3 for PGE$_2$-d9, 353.2/193.3 for PGF$_{2\alpha}$, 576.1/337.1 for Zafirlukast, 423.0/207.0 for Losartan, 272.1/242.1 for Tolcapone, and 329.8/180.85 for Fentiazac, respectively. Analyst software Lab Solution LCMS was used for data manipulation. The uptake rate was normalized with the protein content in cell lysates. The inhibitory effect of the tested compound was expressed as a percentage of control, and the half-maximum inhibitory concentration (IC$_{50}$) was obtained using the following equation with GraphPad Prism10 (GraphPad Software Inc., San Diego, CA):

$$\% \, of \, control = 100 \times \left(1 - \frac{[I]}{IC_{50} + [I]}\right),$$

where [I] is the inhibitor concentration.

## Molecular dynamics simulations

For each cryo-EM model of rat Slco2a1 in complex with a ligand, we conducted conventional all-atom molecular dynamics simulations using GROMACS 2023.1[74] to assess the binding pose stability of the ligand. We embedded the transporter-ligand complex into a pure 1,2-dipalmitoyl-sn-phosphatidylcholine (DPPC) membrane using CHARMM-GUI[75,76], then solvated the complex by adding TIP3P[77] water molecules in a rectangular box with a water thickness of 22.5 Å. Counterions were added to achieve a NaCl concentration of 0.15 M, mimicking physiological ionic conditions. The lipid membrane, transporter, and ligand were respectively parameterized using CHARMM36[78], CHARMM lipid force field[79], and CHARMM General Force Field (CGenFF)[80,81], given their established performance for membrane protein-ligand systems. The protonation states of the system were determined assuming a pH value of 7.4.

To eliminate steric clashes and correct any improper geometry, we performed energy minimization using the steepest descent algorithm until the maximum force in the system dropped below

1000.0 kJ/mol/nm. This was followed by a six-stage equilibration protocol using default parameters from CHARMM-GUI. The protocol comprised two initial NVT equilibration stages using a 1 fs timestep (125 ps in length each) and four subsequent NPT equilibration stages using a 2 fs timestep (125 ps for the first NPT stage and 500 ps for the remaining three). In NPT equilibration, a semi-isotropic stochastic cell rescaling barostat[82] was utilized to maintain the pressure at 1 bar. Throughout the six-stage equilibration, position restraints on the protein backbone, side chains, lipids, and dihedrals were gradually reduced to allow the system to relax while preserving its structural integrity. The temperature was maintained at 310 K during all six stages using a velocity rescaling thermostat[83].

Following equilibration, we performed three replicates of MD simulations without restraints in the NPT ensemble, each for 1 μs. Each replicate was initiated with independent velocity assignments. The same thermostat and barostat settings described above were used. Long-range electrostatic interactions were computed using the Particle Mesh Ewald (PME) method[84] with a cutoff distance of 1.2 nm. Van der Waals interactions were smoothly switched to zero between 1.0 to 1.2 nm. All covalent bonds involving hydrogen atoms were constrained using the LINCS algorithm[85]. The same interaction parameters were also applied during the equilibration stages.

### Analysis of MD simulations

For each replicate ($n = 3$), we computed the root-mean-square deviation (r.m.s.d.) of the ligand heavy atoms over time, with the trajectory aligned to the initial structure using protein backbone atoms to eliminate global rotation and translation. To identify the dominant binding poses, we pooled all three replicates and clustered the binding region, defined as the combination of ligand heavy atoms and the backbone atoms of any residues within 6 Å of the ligand in the cryo-EM structure. We then employed the GROMOS clustering algorithm[86] with an r.m.s.d. cutoff of 1.3 Å to determine the clusters. We performed interaction analysis using ProLIF[87] on frames subsampled every 1 ns from the combined trajectory.

### Reporting summary

Further information on research design is available in the Nature Portfolio Reporting Summary linked to this article.

## Data availability

Atomic coordinates for rat Slco2a1 have been deposited in the Protein Data Bank under accession codes 9QZO (Apo) [https://doi.org/10.2210/pdb9QZO/pdb]; 9R0M (PGE₂) [https://doi.org/10.2210/pdb9R0M/pdb]; 9R0N (PGF₂α) [https://doi.org/10.2210/pdb9R0N/pdb]; 9R0I (Losartan) [https://doi.org/10.2210/pdb9R0I/pdb]; 9R07 (Zafirlukast) [https://doi.org/10.2210/pdb9R07/pdb]; 9R0A (Fentiazac) [https://doi.org/10.2210/pdb9R0A/pdb]; 9R0O (Tolcapone) [https://doi.org/10.2210/pdb9R0O/pdb]. The cryo-EM maps have been deposited in the Electron Microscopy Data Bank (EMDB) under accession codes: EMD-53472 (Apo) EMD-53484 (PGE₂). EMD-53485 (PGF₂α). EMD-53483 (Losartan). EMD-53481 (Zafirlukast). EMD-53482 (Fentiazac). EMD-53486 (Tolcapone). Source data are provided as a Source Data file. Source data are provided with this paper.

## Code availability

The scripts for MD simulation analysis are available in the following GitHub repo: https://github.com/weitse-hsu/SLCO2A1_analysis, and the processed trajectories have been uploaded to Zenodo as entry 17405374

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

## Acknowledgements

This research was supported by Wellcome awards (215519/Z/19/Z & 219531/Z/19/Z) to SN and UKRI BBSRC award BB/Z517215/1 to JLP and SN. Computing was supported via the Advanced Research Computing facility, Oxford, the EPSRC ARCHER2 UK National Supercomputing Service and JADE (*EP/X035603/1*) granted via the High-End Computing Consortium for Biomolecular Simulation (HECBioSim - http://www.hecbiosim.ac.uk), supported by EPSRC (EP/X035603/1) to PCB. This research was funded (in part) by the Intramural Research Program of the NIH to SML. This research was partially conducted with the support of Grant-in-Aids for Scientific Research (KAKENHI) from the Japan Society for the Promotion of Science (24K09819) to TN and (22K15295) to YN, a Research Exchange Grant from Takasaki University of Health and Welfare to TN and YN, and a scholarship donation from Japan Tobacco Inc. (Tokyo, Japan) and a research grant from the Hoansha Foundation (Osaka, Japan) to TN.

## Author contributions

T.K., Y.N., J.L.P., T.N. and S.N. conceived the project. C.J., T.K. and J.D.G. performed all cloning and protein preparation. J.C.D., S.M.L. and C.J. performed all cryo-EM sample processing, data collection and image analysis. J.C.D., C.J., S.M.L. and S.N. constructed the atomic models. W.T.H. and P.C.B. performed all molecular dynamics simulations and analysis. S.N. wrote the manuscript and prepared figures with contributions and discussions from all the authors. The authors wish to thank Dr J Caesar for help in preparing the cryo-EM processing figures.

## Competing interests

The authors declare no competing interests.
