## [Transparent Peer Review file · Nature Communications]

Structural basis for prostaglandin and drug transport via SLCO2A1.

Corresponding Author: Professor Simon Newstead

Version 0:

Reviewer comments:

Reviewer #1

(Remarks to the Author)

The manuscript by Joshi et al. reports the cryo-EM structures of the SLCO2A1 prostaglandin transporter with multiple ligands. By combining structural characterization with functional studies, the authors aim to understand ligand recognition in this important family of transporters.

The biochemical and structural work is solid, and the structures have the potential to provide valuable insights to the field. However, I have significant concerns regarding data interpretation, particularly because the resolution of the maps does not, in my opinion, allow for the detailed analysis proposed by the authors.

PG-bound Structures

While the maps convincingly resolve the protein core (except for the loops and most of TM8), the ligand density is weak. Analysis of the maps shows that one needs to increase the contour level substantially—such that most of the micelle appears, along with various lipid densities—before some density becomes visible in the center of the TM bundle. It is unclear how the authors modeled the prostaglandins based on this limited density. They should discuss the differences in signal between the apo and PG-incubated maps, and the limitations of ligand density.

While partial occupancy could explain the relatively low signal (if the ligand is indeed present), it is not clear on what basis the authors propose a specific binding pose for the PGs or define a precise binding pocket on the protein.

The authors must elaborate on their approach to justify the proposed ligand poses. Even if one accepts that ligand binding is supported by the structure, the definition of the binding pocket—central to the manuscript—is problematic, as few of the proposed residues appear to directly contact the ligand:

- S338 is the closest residue (~2.7 Å), forming a hydrogen bond with the carboxyl group. However, its mutation to alanine does not impair transport (according to Fig. 1d), even though the main text claims otherwise.
- R561, modeled as pointing toward the PG, has poorly resolved side-chain density. The guanidinium group is located more than 4 Å from the E-ring oxygen atoms, which is too far for a polar interaction.
- The authors claim F557 is part of the binding site, supported by loss-of-function in the F557A mutant. Yet this side chain is positioned far from the ligand. Fig. 1C is misleading in this regard. It also suggests a hydrogen bond between S345 and the ω -chain OH group, but the heavy atoms are 4 Å apart.
- M380 is described as important; however, its mutation impacts activity in the human but not the rat transporter (Fig. S4C).

Given these points, I find it difficult to reconcile the functional data with the proposed binding site. On a related note, were the functional assays normalized for transporter expression (as opposed to total protein)? I did not find this information.

In summary, I do not believe the presented data convincingly support the proposed binding pose for prostaglandins.

Structures with Non-PG Ligands

The rationale for ligand selection is somewhat unclear. Some compounds originate from the authors' 2017 screening of FDA-approved drugs, yet others with potential interest (e.g., suramin) were not explored. The physiological relevance of the selected ligands should be clarified.

Regarding the structures: the map resolution appears insufficient to support reliable ligand modeling. While some density is

visible in the TM bundle at high contour levels, the details are insufficient to justify the proposed poses.

In fact, the apparent ligand densities appear at contour levels that reveal annular lipids, which the authors did not model. It is plausible that the central density could also correspond to a lipid, especially in the absence of defined polar interactions.

As an exercise, I reversed the orientation of losartan and could fit it into the density equally well (at least without a quantifiable metric). The same applies to fentiazac. For the latter, given the molecule's symmetry, on what basis do the authors claim that the chloride is facing the intracellular side rather than the extracellular side?

The authors must provide stronger evidence for the proposed poses—beyond weak density maps—before discussing binding pockets, mechanisms, or distinctions between transport and inhibition.

Apo Structure

It is counterintuitive to describe the apo structure at the end of the manuscript, as this should serve as a baseline for comparison. Moreover, the apo structure is referenced earlier in the text. I could not find this structure in the provided structural package.

Manuscript Quality

The manuscript seems rushed (perhaps due to recent depositions of similar structures), which makes it difficult to read in places. For example, lines 71–72 closely repeat lines 67–68.

Table S1 reports an apo structure at 2.9 Å resolution (PDB 9QZO), but I was unable to access this structure.

Overall Recommendation

The study has the potential to be very interesting. However, I am convinced that the structural data do not support the authors' conclusions, likely due to the limitations imposed by the map resolution.

Reviewer #2

(Remarks to the Author)

This work by Joshi et al. is intended to reveal the structural mechanism of SLCO2A1, a key transporter that mediates prostaglandin uptake and clearance. The authors present seven cryo-EM structures of rat SLCO2A1, in the apo state, bound to two prostaglandins (PGE₂ and PGF₂α), and bound to four drugs, at resolutions ranging from 2.9 to 3.4 Å. These structural data are complemented by molecular dynamics simulations and LC-MS/MS-based uptake and inhibition assays. The findings advance our understanding of SLCO2A1 transport and pharmacology. Overall, this is an important study that puts forward exciting hypotheses for future investigation. I would recommend its publication after the following minor points are discussed/addressed.

1. Fig. 1d and Fig. S4c: The human M380A mutant significantly reduces activity, whereas the equivalent residue in rat (M379A) does not. Conversely, the human S339A mutant does not affect activity, while the equivalent residue in rat (S338A) significantly increases activity. It would be helpful if the authors could discuss the potential reasons for these discrepancies.
2. Line 313, Fig. 4c: The density of Tolcapone does not fit the modeled pose well. This should be stated explicitly to ensure transparency for the reader.
3. Line 382: "in the PGF₂α structure, Arg561 points down towards the C1-carboxylate of the substrate, while Glu78 points away towards the extracellular gate and interacts with Lys53 on TM1, breaking their interaction." There appears to be some density around the Glu78 side chain in the EMD-53485 sharpened map, but it seems that this density could be modeled as Ser75 adopting an alternate rotamer. If that is the case, Glu78 would adopt a rotamer similar to the one observed in the PGE₂-bound structure. While it is not uncommon for Glu side chains to be unresolved in cryo-EM maps, and the authors have reasonable grounds to propose this Glu78 movement model, the fact that the Glu78 density is suboptimal should be made explicit.
4. Line 226: Since only a 30-second time point and a single concentration were used in the mass spectrometry assay, is it possible that transport of Fentiazac and Tolcapone is very slow, or that the K_m is high, making uptake undetectable under these conditions? The authors may wish to moderate their claim that these two drugs are not transported.
5. Extra density is present inside the transporter in the putative apo EMD-53472 sharpened map (see screenshots below, between Y223 and around T346). Since the apo structure has not been uploaded (9R0O is uploaded twice), I am unable to assess the model. However, this density does not appear to be CHS. Could the authors comment on what this density might represent and how it may or may not impact the interpretation of the transport mechanism?
6. Lines 66–71: The sentence beginning "There are four types of PGs..." is duplicated.
7. Line 230: "tread PD" should be corrected to "treat PD".
8. Line 231: "detectible" should be corrected to "detectable".
9. Methods section: "ab intio" should be corrected to "ab initio".
10. Fig S2 legend: "plotFig." should be corrected to "plot". Similar corrections are needed in Figs S3 and S8.
11. Figures S2, S3, S5, S6, S8, and S9: It should be formatted as Slco2a1, not SLCO2A1.
12. Line 253: "Ile 344" should read "Leu 344".

Chia-Hsueh Lee

Reviewer #3

(Remarks to the Author)

Reviewer #4

(Remarks to the Author)

The manuscript entitled “Structural basis for prostaglandin and drug transport via SLC2OA1” describes the three-dimensional structures and simulations of the wild-type rat SlcO2a1 transporter in complex with endogenous prostaglandins, transported drugs, and drug inhibitors. The work performed by Joshi, Deme, Nakamura, and colleagues is crucial for understanding the transport mechanism and selectivity of SLC2OA1, a protein responsible for the distribution of endogenous and exogenous molecules throughout our body. The authors propose a simple mechanistic model and key residues involved in substrate binding and translocation, comparing it to the structural determinants described for SLCO1B1 and SLCO1B3.

The study is robust, but I have a few comments regarding the analyses of the SLC2OA1-ligands interactions and the mechanistic model proposed:

- 1) The authors' choice to name the human transporter SLC2OA1 and the rat transporter SlcO2a1 confused me. In my opinion, using the prefixes h for human (hSLC2OA1) and r for rat (rSLC2OA1) could improve the clarity of the manuscript.
- 2) The clustering performed provides useful insight into the stability of the ligand pose. However, time-series plots of the protein-ligands interactions give a better representation and quantification of the stability. Since the interactions have been analyzed with ProLIF (I guess ProLI is a typo at line 793), it will be quite easy to produce these plots that can be added as supplementary figures.
- 3) Moreover, ProLIF allows you to analyze salt bridges, pi-stacking, and cation-pi interactions. This information is crucial to understanding if and how the ligands (prostaglandins and drugs) interact with Trp565 and Arg561. Please add these interactions, if relevant, to interaction plots (Figure 2C and 2D) and time series.
- 4) At lines 333-334, the authors state that: “Zafirlukast, and Losartan, neither molecule appears capable of productively engaging Trp565, the hydrophobic band, and positioning a negative group in the C1-carboxyl binding site, which, as discussed below, are likely to form critical requirements that trigger transport in SLCO2A1.”. However, Zafirlukast, and Losartan have been classified as translocated ligands. Is it a typo?
- 5) I want to highlight that only Losartan appears to bind in proximity to Trp565. Therefore, experimental evidence does not fully support the authors' claim about the importance of engaging Trp565 to trigger transport. Please clarify.
- 6) The authors could better support the interactions identified for Zafirlukast, Losartan, Fentiazac, and Tolcapone with MD simulations, repeating what they have done for prostaglandin-bound structures. This would allow for more meaningful comparisons of interaction patterns and stability.
- 7) The authors repeatedly state that Arg56-Glu78 salt bridge has a crucial role in the protein mechanism. They also suggest that the inability of compounds to stabilize this interaction is responsible for the inhibitory binding of Fentiazac and Tolcapone. This claim could be further supported by a more extensive analysis of this interaction during the MD simulations of the protein-ligands bound systems.
- 8) The authors could support the theory of the “PG entrance from the membrane” with some simulations. For instance, MD simulations of the CHS-bound structure could give indications regarding the stability of the modelled molecule. This will add important information that could support the alternating access transport model proposed.
- 9) In the Methods section, Molecular Dynamics Simulations paragraph, please indicate the GROMACS version you used.

As a final remark, I would like to mention that it was difficult to analyze the results without having access to the structural models, trajectory snapshots, or analysis scripts. Please consider providing that for reviewers and readers, and take into account the Reliability and reproducibility checklist for molecular dynamics simulations <https://www.nature.com/articles/s42003-023-04653-0> included in the Nature Communication Computational tools reporting guidelines <https://www.nature.com/ncomms/submit/resources>.

Version 1:

Reviewer comments:

Reviewer #1

(Remarks to the Author)

-The authors propose that the binding site for the E-ring of the ligand is achieved by a “positive density” created by R561 and “helped” by W565 and F557.

R561 in the apo structure Hia et al or Yu et al, adopt a similar conformation as the one proposed here, suggesting that either something such as another sidechain (see below) or an ion must stabilize the side chain in absence of ligand (or that the Arg is not protonated but that is less likely).

On the other hand, Xia et al propose a different rotamer for R561 in the PG-bound state, pointing away from the ligand (these authors point this absence of direct interaction).

This Arg is conserved in the OATP family. Do the authors suggest that this is conserved “positive” patch that allows for binding of anionic substrates throughout the family? This would clearly not be supported by the structures of OATP1C1 in

complex with various ligand. In fact, W565 is not conserved and all while F557 is partly conserved (often changed for a Thr). In my opinion, analysis of the structures and sequence conservation throughout the global family does not support a role for W561 in ligand binding but rather as a key element in the structure, possibly as a hinge during conformational transition. A likely partner is the highly conserved E78, which forms a salt bridge with the conserved Arg, as observed in the structures of OATPB1 and OATP1C1.

In other terms I believe that the proposed role of R561 proposed by the authors is not supported neither by their structure or the literature.

-To my original critique "S338 is the closest residue (~2.7 Å), forming a hydrogen bond with the carboxyl group. However, its mutation to alanine does not impair transport (according to Fig. 1d), even though the main text claims otherwise", the authors respond: "This has now been addressed in lines 273-277." There is no such mention in lines 273-277.

-

Similarly, the authors reply :

"Our hypothesis is that the close position of Arg561 to Phe557 and Trp565, along with the predominantly hydrophobic nature of the binding site, creates a localised positive patch of electrostatic density that attracts the E-ring and C1-Carboxylate of the PG ligand. We have detailed our reasoning in lines 225-235."

But I did not find such discussion at those lines.

-The author did not address my point on Fig 1C being misleading regarding the H-bond with S345

Reviewer #2

(Remarks to the Author)

The authors have addressed my questions satisfactorily.

One minor issue:

Supplemental Fig. 7d: The figure shows Losartan interactions, not Tolcapone interactions. Please address this inconsistency between the figure and the figure legend.

Reviewer #3

(Remarks to the Author)

Reviewer #4

(Remarks to the Author)

I would like to thank the authors for revising the manuscript and responding to my comments. I don't have any other concern.

Version 2:

Reviewer comments:

Reviewer #1

(Remarks to the Author)

In their latest rebuttal, the authors kindly educate me on the difference between ligand recognition and transport and suggest that I misinterpreted their model where ligand recognition (involving W565), will disrupt the R561-E78 bridge, the latter being a conserved mechanism in the SLCO family.

Nevertheless, the manuscript states (line 153 in the latest version):

"Substitution of Arg561 with alanine abolished transport. Replacement with either lysine, glutamine or leucine reduced activity in human SLCO2A1 to <20% of WT levels, demonstrating the critical importance of the positively charged guanidino group for PG recognition."

It is unclear how this sentence should be interpreted if not as evidence that R561 is directly involved in ligand recognition.

The underlying question concerns the relationship between ligand binding/recognition and transport (i.e. through disruption of the R561-E78 interaction). My understanding is that the authors propose that the anionic charge of the ligand perturbs this interaction, thereby directly linking binding to the transport mechanism. This is a plausible model; however, I note that:

a) In both the apo (9QZO) and PG-bound (9R0M) structures, formation of an R561-E78 salt bridge appears unlikely, given the relatively long heavy-atom distances (4.5 Å and 3.5 Å, respectively) and geometries incompatible with a canonical salt bridge.

b) The interaction is observed in both apo and PG-bound structures reported by Xia et al.

Based on this information, I maintain my position regarding the points raised above. That said, the authors are naturally best placed to interpret their own data and are entitled to propose interpretations, models, and hypotheses.

While scientific dialogue between authors and referees is a critical component of peer review, interesting and well-executed studies such as this one must be published at some point. I therefore recommend publication of the study.

We appreciate and value the additional workload involved in reviewing our paper and thank the reviewers for their time and feedback. Below, we outline our changes to the manuscript and provide explanations of the experimental design and conclusions. Amendments to the main text are highlighted in green to help identify key updates.

Reviewer 1.

The biochemical and structural work is solid, and the structures have the potential to provide valuable insights to the field. However, I have significant concerns regarding data interpretation, particularly because the resolution of the maps does not, in my opinion, allow for the detailed analysis proposed by the authors.

We appreciate the time taken to review our data, and we hope to have addressed the reviewer's concerns in our response, explanations, and additional MD data.

While partial occupancy could explain the relatively low signal (if the ligand is indeed present), it is not clear on what basis the authors propose a specific binding pose for the PGs or define a precise binding pocket on the protein.

We modelled the PG ligands based on EM volumes that show a bifurcation from a central point near Trp565 (Fig. 1a-b). After examining the electrostatic properties of the binding site, we decided to model the PG ligands with the C1-Carboxylate facing the positive density created by Arg561, which is close to Phe557 and Trp565. As we discuss later, our hypothesis is that this positive density in the binding site plays a key role in ligand recognition. While our study was under revision, a further publication reporting the structure of human SLCO2A1 was published (Xie et al., Nature Communications, 2025). The two structures agree very well with each other (r.m.s.d. between the two structures is 0.79 Å over 495 Ca atoms), and both show the outward open state of the transporter. We were pleased to see that our modelled PGE₂ and PGF_{2α} are in the same position as the one modelled for PGE₂ in this independent study (Fig. 1c). The authors of this study also noted the lack of extensive interactions between the PGE₂ molecule and the transporter and identified only one hydrogen bond to the C1-carboxylate, via Ser339 (Ser338 in the rat homologue). The conclusion from this study, as in ours, is that the main interactions with the PG molecule are via van der Waals and hydrophobic interactions, with Arg561 orienting the ligand via the E-ring and the C1-carboxylate.

Figure 1. Analysis of Cryo-EM volume for PGE₂. **a**, Schematic of PGE₂. **b**, Cryo-EM volume density for PGE₂ is shown (orange and threshold 0.175). **c**, Structural overlay of the human

SLCO2A1 (PDB: 9JV1; grey) with the structure of the rat Slco2a1 (PDB: 9R0M; this study and coloured).

The site-directed mutagenesis data in the Xie et al. study also agree well with those reported in our paper, except for the effect of Ser339Ala. In the Xie et al. study, this mutation, along with Ser339Pro, abolished transport activity in SLCO2A1, whereas in our study, we did not observe a noticeable impact. We suggest this discrepancy may be due to the different ways in which our studies measured transport function. The Xie et al. study used 6-carboxyfluorescein (6-CF) as a reporter ligand, whereas our study employed direct detection by LC-MS/MS. However, the Xie et al. study does support the role of Ser339 in SLCO2A1 transport with 6-CF.

We have now amended our paper to bring this new work to the reader's attention and to discuss its implications for understanding the mechanism of PG transport via SLCO2A1. Please see lines: 185-186.

Our new explanation for why we modelled the PG molecules in the positions described above is detailed in lines 225-235.

We have reworded our description of the mutagenesis results to correct our error in describing the Ser339Ala mutation and to include the new data on Ser346Ala, which correlate nicely with the Xie et al. study. Now detailed in lines 273-277.

• S338 is the closest residue (~2.7 Å), forming a hydrogen bond with the carboxyl group. However, its mutation to alanine does not impair transport (according to Fig. 1d), even though the main text claims otherwise.

This has now been addressed in lines 273-277.

• R561, modeled as pointing toward the PG, has poorly resolved side-chain density. The guanidinium group is located more than 4 Å from the E-ring oxygen atoms, which is too far for a polar interaction.

Our hypothesis is that the close position of Arg561 to Phe557 and Trp565, along with the predominantly hydrophobic nature of the binding site, creates a localised positive patch of electrostatic density that attracts the E-ring and C1-Carboxylate of the PG ligand. We have detailed our reasoning in lines 225-235.

• The authors claim F557 is part of the binding site, supported by loss-of-function in the F557A mutant. Yet this side chain is positioned far from the ligand. Fig. 1C is misleading in this regard. It also suggests a hydrogen bond between S345 and the ω-chain OH group, but the heavy atoms are 4 Å apart.

As discussed above, our hypothesis is that Phe557 modulates or amplifies the positive electrostatic potential generated by Arg561, thereby assisting ligand recognition. This idea is supported by our assay data, which shows that only an aromatic side chain retains function, while a hydrophobic alanine results in loss of activity.

- M380 is described as important; however, its mutation impacts activity in the human but not the rat transporter (Fig. S4C).

We maintain our hypothesis that Met380 is important based on our assay data and the close proximity of this side chain to the PG ligands. In addition, our molecular dynamics data also highlight the importance of van der Waals and hydrophobic interactions with the ligand. In the newly added time series plots (Supplementary Fig. 4e-f), you can see that Met379 remains in contact with both PGE₂ and PGF_{2α} throughout the 1μs simulation.

On a related note, were the functional assays normalized for transporter expression (as opposed to total protein)? I did not find this information.

We apologise for not including this data in our original submission. This data is now included as additional data in the submission and repeated below. The total protein of each mutant expressed in HEK cells was analysed by Western blotting, and the degree of expression was quantitatively determined using appropriate software. The extent of expression of each mutant was compared to that of WT cells in the same blot, and the relative ratio of the expression was reflected in the result of PGE₂ uptake.

The rationale for ligand selection is somewhat unclear. Some compounds originate from the authors' 2017 screening of FDA-approved drugs, yet others with potential interest (e.g., suramin) were not explored. The physiological relevance of the selected ligands should be clarified.

It was difficult to select drugs because little information is available on the interaction of SLCO2A1 and drugs. We chose Losartan and Zafirlukast as an excellent choice among the few known substrates of SLCO2A1, and Fentiazac and Tolcapone as potential inhibitors, which we confirm in our study.

Regarding the structures, the map resolution appears insufficient to support reliable ligand modelling. While some density is visible in the TM bundle at high contour levels, the details are insufficient to justify the proposed poses. In fact, the apparent ligand densities appear at contour levels that reveal annular lipids, which the authors did not model. It is plausible that the central density could also correspond to a lipid, especially in the absence of defined polar interactions.

We did not observe density in the binding site of the apo structure in the same position as that obtained for the ligands. In the apo structure, we did observe density for a cholesterol hemisuccinate molecule, as detailed in Fig. 6 and lines 831-883. However, we are confident that the densities we assign to the drugs are not those of lipid molecules.

To further address this comment, we have conducted additional MD simulations on the drug-bound structures using the same methodology applied to the PG molecules. These are now detailed in the relevant sections of the study 'Structural basis for drug recognition by SLCO2A1' and 'Structural basis for inhibitor recognition'. In brief, the MD results demonstrate that Losartan and Zafirlukast have stabilities comparable to those of PGE2 and PGF2a at the binding site, whereas Fentiazac and Tolcapone are less stable. We have incorporated the new MD data and analyses to conclude that Fentiazac and Tolcapone are unable to trigger transport because they cannot adopt stable binding poses, in contrast to Losartan and Zafirlukast.

As an exercise, I reversed the orientation of losartan and could fit it into the density equally well (at least without a quantifiable metric). The same applies to fentiazac. For the latter, given the molecule's symmetry, on what basis do the authors claim that the chloride is facing the intracellular side rather than the extracellular side?

As the reviewer can well appreciate, the pseudo-symmetric structure of Fentiazac and

Tolcapone made it challenging to model these ligands into the cryo-EM volumes. However, we have explained our reasoning for our decision in the paper as follows: “The pseudo-symmetrical structure of the compound made it challenging to model the orientation of the drug, and we leave open the possibility that the drug might also engage in the opposite orientation. However, using our prior observations from the PG-bound structures, we modelled the drug with the chloride atom occupying a similar position to the carboxylates, which we termed the negative cluster”. We also modelled Tolcapone in the same orientation as we did for the PG molecules, based on our observations.

The authors must provide stronger evidence for the proposed poses—beyond weak density maps—before discussing binding pockets, mechanisms, or distinctions between transport and inhibition.

We agree and, in response to this reasonable request, have provided the additional MD simulations and their associated analyses in the revised manuscript.

Apo Structure

It is counterintuitive to describe the apo structure at the end of the manuscript, as this should serve as a baseline for comparison. Moreover, the apo structure is referenced earlier in the text. I could not find this structure in the provided structural package.

We apologise for the mix-up during submission; this was due to exasperation with the FigShare file server. The decision to introduce the apo structure at the end of the study was based on the study's narrative, which aimed to understand how SLCO2A1 recognises prostaglandins and drug molecules, rather than simply determining its 3D structure. Indeed, our study followed the timeline reported in the paper, with the apo structure as the last structure we obtained. The apo structure was very helpful in understanding how PG binding might trigger transport (by interacting with Arg561), which was a section detailed towards the end of the study. We therefore chose the somewhat unconventional approach to describe the ligand-bound structures first.

Reviewer 2.

The findings advance our understanding of SLCO2A1 transport and pharmacology. Overall, this is an important study that puts forward exciting hypotheses for future investigation. I would recommend its publication after the following minor points are discussed/addressed.

We thank the reviewer for their support and contribution to our study.

1. Fig. 1d and Fig. S4c: The human M380A mutant significantly reduces activity, whereas the equivalent residue in rat (M379A) does not. Conversely, the human S339A mutant does not affect activity, while the equivalent residue in rat (S338A) significantly increases activity. It would be helpful if the authors could discuss the potential reasons for these discrepancies.

It was surprising to observe species-specific differences in the SLCO2A1 transporters between rats and humans. In truth, we are not sure why the rat transport behaves differently from the human homologue. However, since the focus of this study is understanding the human transporter, we felt that delving into a detailed analysis of why these two proteins behave differently with respect to these mutations might cause confusion. The recent publication by Xie et al. also supports our conclusions about the human transporter (with the notable exception of Ser346 discussed above).

2. Line 313, Fig. 4c: The density of Tolcapone does not fit the modeled pose well. This should be stated explicitly to ensure transparency for the reader.

We have now included a detailed MD analysis of the ligand binding poses and included the following rationale for our modelled pose in lines 760-849. “In the Tolcapone bound structure, we again suffered from limited resolution and pseudo-symmetry in the molecule. Using the same rationale as we applied to the Fentiazac compound, we modelled the nitro group near the α -chain of the PG substrates, and the carbonyl group occupies the same position as the C15-hydroxyl group on the ω -chain of PGE₂ (Fig. 4d).”

3. Line 382: "in the PGF₂ α structure, Arg561 points down towards the C1-carboxylate of the substrate, while Glu78 points away towards the extracellular gate and interacts with Lys53 on TM1, breaking their interaction." There appears to be some density around the Glu78 side chain in the EMD-53485 sharpened map, but it seems that this density could be modeled as Ser75 adopting an alternate rotamer. If that is the case, Glu78 would adopt a rotamer similar to the one observed in the PGE₂-bound structure. While it is not uncommon for Glu side chains to be unresolved in cryo-EM maps, and the authors have reasonable grounds to propose this Glu78 movement model, the fact that the Glu78 density is suboptimal should be made explicit.

We thank the referee for pointing this out, and we have amended the text to bring this alternative suggestion to their attention, as detailed: “However, in the PGF_{2 α} structure, Arg561 points down towards the C1-carboxylate of the substrate, while Glu78 points away towards the extracellular gate and interacts with Lys53 on TM1, breaking their interaction. However, the density around Glu78 is not unambiguous, and it is possible Arg561 could move away without Glu78 needing to adopt the alternate rotamer”.

4. Line 226: Since only a 30-second time point and a single concentration were used in the mass spectrometry assay, is it possible that transport of Fentiazac and Tolcapone is very slow, or that the Km is high, making uptake undetectable under these conditions? The authors may wish to moderate their claim that these two drugs are not transported.

Unfortunately, we have not yet optimised the conditions for the uptake of Fentiazac and Tolcapone, for which SLCO2A1-mediated uptake is detectable, although different time points, substrate concentrations, and the effect of BSA (to prevent interaction of these drugs with proteins or the membrane surface) have been studied. The main problem is their high hydrophilicity; the expected logP values for tolcapone and fentiazac are 3.06 and 5, respectively, which means they diffuse quickly across plasma membranes. As a result, the contribution of SLCO2A1 to their uptake is too low to be measured. Given the limitations listed above, we have taken the reviewer’s suggestion and added a caveat that, under our assay conditions, Fentiazac and Tolcapone are likely to act as inhibitors.

5. Extra density is present inside the transporter in the putative apo EMD-53472 sharpened map (see screenshots below, between Y223 and around T346). Since the apo structure has not been uploaded (9R00 is uploaded twice), I am unable to assess the model. However, this density does not appear to be CHS. Could the authors comment on what this density might represent and how it may or may not impact the interpretation of the transport mechanism?

We had previously brought this density to the attention of the reader as a ‘lipid-like’ density in the lateral opening. However, we have now modelled cholesterol hemisuccinate in the density and explicitly identified it in the text. Line 956 “In our cryo-EM maps for the apo and ligand-bound

structures, we observe a concentration of lipid-like densities clustering within this opening, specifically between TM5 from the N-terminal bundle and TM8 from the C-terminal bundle”.

6. Lines 66–71: The sentence beginning “There are four types of PGs...” is duplicated.

Thank you, and corrected.

7. Line 230: “tread PD” should be corrected to “treat PD”.

Thank you, and corrected.

8. Line 231: “detectible” should be corrected to “detectable”.

Thank you, and corrected.

9. Methods section: “ab intio” should be corrected to “ab initio”.

Thank you, and corrected.

10. Fig S2 legend: “plotFig.” should be corrected to “plot”. Similar corrections are needed in Figs S3 and S8.

Thank you, and corrected.

11. Figures S2, S3, S5, S6, S8, and S9: It should be formatted as *Slco2a1*, not *SLCO2A1*.

Thank you, and corrected.

12. Line 253: “Ile 344” should read “Leu 344”.

Thank you, and corrected.

Reviewer 4.

The study is robust, but I have a few comments regarding the analyses of the SLC2OA1-ligands interactions and the mechanistic model proposed:

We thank the reviewer for their support and the time taken to read and contribute to our work.

1) The authors’ choice to name the human transporter SLC2OA1 and the rat transporter Slco2a1 confused me. In my opinion, using the prefixes h for human (hSLC2OA1) and r for rat (rSLC2OA1) could improve the clarity of the manuscript.

In the original manuscript, we adhered to the convention in the OATP field of referring to the human transporter as all capital letters (SLCO2A1) and the rat homologue as lowercase letters (Slco2a1). However, we recognise that this distinction can be confusing for a general audience. To clarify, we have included ‘human’ and ‘rat’ throughout the text to make it clear which protein we are discussing.

2) The clustering performed provides useful insight into the stability of the ligand pose. However, time-series plots of the protein-ligands interactions give a better representation and quantification of the stability. Since the interactions have been analyzed with ProLIF (I guess ProLI is a typo at line 793), it will be quite easy to produce these plots that can be added as supplementary figures.

We agree with the reviewer, and during the revision, we have added these plots to the Supplementary Figures and included their analysis in the study. We have also undertaken additional MD simulations on the remaining ligands reported in our first draft and extensively rewrote sections of the paper to incorporate this new data and analysis.

3) Moreover, ProLIF allows you to analyze salt bridges, pi-stacking, and cation-pi interactions. This information is crucial to understanding if and how the ligands (prostaglandins and drugs) interact with Trp565 and Arg561. Please add these interactions, if relevant, to interaction plots (Figure 2C and 2D) and time series.

We agree and have included the interaction frequency plots, along with time series plots for each ligand, in the revised paper, and we discuss the insights obtained in the manuscript.

4) At lines 333-334, the authors state that: “Zafirlukast, and Losartan, neither molecule appears capable of productively engaging Trp565, the hydrophobic band, and positioning a negative group in the C1-carboxyl binding site, which, as discussed below, are likely to form critical requirements that trigger transport in SLCO2A1.” However, Zafirlukast, and Losartan have been classified as translocated ligands. Is it a typo?

Thank you, we have rewritten this section for clarity.” However, unlike PGE₂, PGF_{2α}, Zafirlukast, and Losartan, neither Fentiazac and Tolcapone molecule seems capable of adopting a stable binding pose in the transporter and effectively engaging Arg561, and disrupting the salt bridge networks that, as discussed below, control the conformational changes that facilitate transport”.

5) I want to highlight that only Losartan appears to bind in proximity to Trp565. Therefore, experimental evidence does not fully support the authors' claim about the importance of engaging Trp565 to trigger transport. Please clarify.

Trp565 is one of the few changes in the binding site of SLCO2A1 compared to other SLCO family members that do not transport PG molecules. Therefore, Trp565 likely plays a role in PG transport. This is the basis for our focus on Trp565 in this paper. However, we acknowledge that the interaction may not be direct or stable, as indicated by our MD simulation data. We have therefore revised our language regarding the necessity of engaging Trp565 to initiate transport. Further biochemical, structural, and simulation studies will be required to fully understand the role of Trp565 in PG and drug transport through SLCO2A1. In addition, we have conducted new experiments to investigate the effects of mutating Trp565 and Arg561 on the transport of Zafirlukast and Losartan. The new data, now shown in **Fig. 3g**, reveal that mutating either of these side chains substantially reduces the transport activity for these drugs. The data for Zafirlukast is a little noisy, due to the higher background with this drug. However, we are confident that the data demonstrate a loss of uptake with either Trp565 or Arg561. We therefore included this new data to support our hypothesis.

6) The authors could better support the interactions identified for Zafirlukast, Losartan, Fentiazac, and Tolcapone with MD simulations, repeating what they have done for prostaglandin-bound structures. This would allow for more meaningful comparisons of interaction patterns and stability.

We agree, and during the revision period, we added extensive additional MD simulation data on the drug molecules and discussed their implications for our model of ligand recognition and transport via SLCO2A1.

7) The authors repeatedly state that Arg56-Glu78 salt bridge has a crucial role in the protein mechanism. They also suggest that the inability of compounds to stabilize this interaction is responsible for the inhibitory binding of Fentiazac and Tolcapone. This claim could be further supported by a more extensive analysis of this interaction during the MD simulations of the protein-ligands bound systems.

We agree and have now included the r.m.s.d plots to support our model of the Arg561-Glu78 salt-bridge mechanism. However, we also recognise that our data cannot definitively establish the mechanism we propose. Nevertheless, based on the data presented here from the structures, AlphaFold models, MD simulation data, and mutagenesis effects on PG transport, we are confident that our model provides a strong foundation for further research on this fascinating transporter.

8) The authors could support the theory of the “PG entrance from the membrane” with some simulations. For instance, MD simulations of the CHS-bound structure could give indications regarding the stability of the modelled molecule. This will add important information that could support the alternating access transport model proposed.

We agree. However, we believe that the additional work required to thoroughly investigate this point using MD and biochemical/biophysical methods is beyond the scope of the present study. We are currently working on this issue for another study.

9) In the Methods section, Molecular Dynamics Simulations paragraph, please indicate the GROMACS version you used.

Thank you for highlighting this omission. We have now corrected the methods section. We used GROMACS

As a final remark, I would like to mention that it was difficult to analyze the results without having access to the structural models, trajectory snapshots, or analysis scripts.

Our apologies. We have now uploaded all analysis scripts to GitHub and processed MD trajectories to Zenodo, accessible here: https://github.com/weitse-hsu/SLCO2A1_analysis and <https://doi.org/10.5281/zenodo.17405374>, respectively.

We appreciate the reviewers' feedback, and we have addressed their concerns below.

Reviewer 1.

-The authors propose that the binding site for the E-ring of the ligand is achieved by a “positive density” created by R561 and “helped” by W565 and F557.

R561 in the apo structure Hia et al or Yu et al, adopt a similar conformation as the one proposed here, suggesting that either something such as another sidechain (see below) or an ion must stabilize the side chain in absence of ligand (or that the Arg is not protonated but that is less likely). On the other hand, Xia et al propose a different rotamer for R561 in the PG-bound state, pointing away from the ligand (these authors point this absence of direct interaction). This Arg is conserved in the OATP family. Do the authors suggest that this is conserved “positive” patch that allows for binding of anionic substrates throughout the family? This would clearly not be supported by the structures of OATP1C1 in complex with various ligand. In fact, W565 is not conserved and all while F557 is partly conserved (often changed for a Thr).

In my opinion, analysis of the structures and sequence conservation throughout the global family does not support a role for W561 in ligand binding but rather as a key element in the structure, possibly as a hinge during conformational transition. A likely partner is the highly conserved E78, which forms a salt bridge with the conserved Arg, as observed in the structures of OATPB1 and OATP1C1.

In other terms I believe that the proposed role of R561 proposed by the authors is not supported neither by their structure or the literature.

We believe the reviewer has been confused by the following sentence towards the end of our paper:

Arginine 561 and Glu78 are strictly conserved within the SLCO superfamily and are essential for PG transport^{9,48,54}, with even a conservative substitution of Glu78 to aspartate reducing function by > 60 % (Fig. 5c). The equivalent arginine in SLCO1B1 and SLCO1B3, Arg580 (Supplementary Fig. 1), is also essential for function^{55,56}, similar to our observation in SLCO2A1 (Fig. 1d), suggesting the mechanism we propose may be a universal feature of transport within the SLCO superfamily.

The mechanism we refer to here is the one where the Arg561-Glu78 interaction must be disrupted to facilitate transport.

The reviewer appears to believe we are suggesting that Arg561 is essential for ligand recognition in all members of the OATP superfamily and PG recognition in SLCO2A1.

This is not what this paragraph asserts. We are simply saying that our data and structural analysis suggest that the strictly conserved salt bridge between Arg561 and Glu78 is a universal mechanism by which ligands, once bound, trigger a conformational change from outward to inward open states, resulting in transport **within the wider OATP family**. We believe this is an important finding and idea from our analysis.

Ligand recognition and transport do not have to be the same event, although one must precede the other! In the case of SLCO2A1, electrostatic complementarity with the PG ligands results in recognition, as we discuss in the paper and the reviewer alludes to above. As we suggest and illustrate in Fig. 5, our hypothesis is that the disruption of the Arg561-Glu78 salt bridge forms part of the transport step, not the specific ligand recognition step. Once the anionic ligands are correctly bound, their interaction with the equivalent side chain to Arg561 in SLCO2A1 weakens the salt bridge and triggers transport within the wider SLCO family.

To clarify this point further, we have added the following additional text in green to this section of the paper (lines 495-501).

Arginine 561 and Glu78 are strictly conserved within the SLCO superfamily and are essential for PG transport^{9,48,54}, with even a conservative substitution of Glu78 to aspartate reducing function by > 60 % (Fig. 5c). The equivalent arginine in SLCO1B1 and SLCO1B3, Arg580 (Supplementary Fig. 1), is also essential for function^{55,56}, similar to our observation in SLCO2A1 (Fig. 1d). Our model therefore suggests that ligand recognition and transport may be separate events within the OATP family. Each member of the OATP family has specific side chains that recognise their respective ligands; in the case of SLCO2A1, this would be principally Trp565. Once successfully bound, the ligands would then disrupt the Arg-Glu interaction, triggering transport. This mechanism is an elegant way for the SLCO superfamily to evolve specific ligand recognition mechanisms whilst retaining a conserved mechanism to trigger alternating access transport.

This would clearly not be supported by the structures of OATP1C1 in complex with various ligand.

Actually, we would argue that our mechanism is supported by recent structures of OATP1C1. In the recent study by Ge et al., Cell 2025 DOI: 10.1016/j.cell.2025.06.032, which reports the cryoEM structure of OATP1C1 bound to thyroxine T4, the ligand is recognised by Lys376 but sits within 4.2 Å of Arg597 (equivalent to Arg561 in SLCO2A1). We would argue this structure supports our model, that the anionic group of T4 disrupts the Arg597-Glu89 interaction in OATP1C1. Indeed, it was also previously reported that mutation of Arg597 disrupts ligand recognition and transport (Westholm et al., Endocrinology 2010 DOI: 10.1210/en.2010-0640), which is consistent with our proposed model.

Figure 1. Comparison of ligand binding sites between OATP1C1 and SLCO2A1. **A**, CryoEM structure of human OATP1C1 bound to thyroid hormone T4. The proximity (4.1Å) of the carboxylate on T4 to Arg597 suggests a similar interaction with Arg597 as proposed in our study. **B**, Figure 5d from our paper showing the structural comparison between the different structures of rat SLCO2A1 analysed in our study. Our model suggests that disruption of the Arg-Glu interaction promotes the movement of TM2 to initiate transport.

-To my original critique “S338 is the closest residue (~2.7 Å.), forming a hydrogen bond with the carboxyl group. However, its mutation to alanine does not impair transport (according to Fig. 1d), even though the main text claims otherwise”, the authors respond: “This has now been addressed in lines 273-277.” There is no such mention in lines 273-277.

This was our mistake; it was lines 195-200. I suspect this was due to formatting changes between the rebuttal letter and the final version... The original revised text is below in green:

Alanine substitution of Met379 (Met380 in human SLCO2A1) also reduces PGE₂ transport to ~30% WT levels, demonstrating the importance of this side chain in PG recognition. However, replacement of Ser338 (Ser339 in human SLCO2A1) had little impact on SLCO2A1 activity (Fig. 1d and Supplementary Fig. 4b), suggesting this side chain does not play an important role in PG recognition.

Similarly, the authors reply:

“Our hypothesis is that the close position of Arg561 to Phe557 and Trp565, along with the predominantly hydrophobic nature of the binding site, creates a localised positive patch of electrostatic density that attracts the E-ring and C1-Carboxylate of the PG ligand. We have detailed our reasoning in lines 225-235.”

But I did not find such discussion at those lines.

This was our mistake; it was lines 180-190. I suspect this was due to formatting changes between the rebuttal letter and the final version... The original revised text is below in green:

The carboxylate groups of the α -chain of PGE₂ and PGF_{2 α} , which impart a net negative charge to these molecules at physiological pH, curl beneath their respective E-rings and face the positively charged region created by the Arg561, Phe557, and Trp565 cluster. The position of the carboxylate group in PGE₂ and PGF_{2 α} may also explain why the aromatic groups of Trp565 and Phe557 are important for PG recognition. The proximity of these two aromatic side chains to Arg561 amplifies the positive charge of this side chain within the binding site, which would facilitate the orientation of the E-ring and carboxylate group observed in the cryo-EM structures. As discussed below, Arg561 is also likely to play an essential role in the general alternating access mechanism within the SLCO superfamily, along with its interaction with Glu78 on TM5.

-The author did not address my point on Fig. 1C being misleading regarding the H-bond with S345

We disagree that this figure is misleading for the following reasons. The figure shows a schematic of the binding site and the residues that are in proximity to the ligand. I think anyone would agree that 4 Å is close. We do not suggest these are making a hydrogen bond; however, it is not beyond the realms of imagination to think that, during the Brownian motion of the ligand in the binding site, one might form. In addition, the figure illustrates why we test the impact of the equivalent side chains in the human transporter (S339A and S346A) on PG transport. Given that neither of these side chains had any effect, this indicates their lack of importance for PG recognition. However, omitting them from this figure could lead to confusion among other readers, as they might wonder why we included them in the panel of mutations in Fig. 1d.

Reviewer #2 (Remarks to the Author):

The authors have addressed my questions satisfactorily.

One minor issue:

Supplemental Fig. 7d: The figure shows Losartan interactions, not Tolcapone interactions. Please address this inconsistency between the figure and the figure legend.

We have addressed this in the revised version.

Reviewer #4 (Remarks to the Author):

I would like to thank the authors for revising the manuscript and responding to my comments. I don't have any other concern.